# Fat body-derived cytokine Upd2 controls disciplined migration of tracheal stem cells in *Drosophila*

**Pengzhen Dong[1,2], Yue Li[1,2,3], Yuying Wang[1,2], Qiang Zhao[1,2], Tianfeng Lu[4], Jian Chen[1,2], Tianyu Guo[1,2], Jun Ma[5,6], Bing Yang[7]\*, Honggang Wu[3]\*, Hai Huang[1,2,8]\***

[1]Second Affiliated Hospital, and Department of Cell Biology, Zhejiang University School of Medicine, Hangzhou, China; [2]State Key Laboratory of Transvascular Implantation Devices, Hangzhou, China; [3]Zhejiang Key Laboratory of Precision Diagnosis and Therapy for Major Gynecological Diseases, Women's Hospital, Zhejiang University School of Medicine, Hangzhou, China; [4]Department of Developmental Biology and Neuroscience, Washington University in St. Louis, Missouri, Washington DC, United States; [5]Center for Genetic Medicine, the Fourth Affiliated Hospital, Zhejiang University, School of Medicine, Hangzhou, China; [6]Institute of Genetics, Zhejiang University International School of Medicine, Hangzhou, China; [7]MOE Laboratory of Biosystem Homeostasis and Protection and Life Sciences, Institute, Zhejiang University, Hangzhou, China; [8]Zhejiang Provincial Key Laboratory of Genetic and Developmental Disorders, Zhejiang University School of Medicin, Hangzhou, China

**\*For correspondence:**
bingyang@zju.edu.cn (BY);
honggangwu@zju.edu.cn (HW);
haihuang@zju.edu.cn (HH)

## eLife Assessment

This **valuable** study investigates how inter-organ communication between the tracheal stem cells and the fat body plays a key role in the directed migration of tracheal stem cells in *Drosophila* pupae. The evidence supporting the conclusions is **convincing**. The work would be of broad interest to researchers in the fields of developmental biology and cancer biology.

**Abstract** Coordinated activation and directional migration of adult stem cells are essential for maintaining tissue homeostasis. *Drosophila* tracheal progenitors are adult stem cells that migrate posteriorly along the dorsal trunk to replenish degenerating branches that disperse the fibroblast growth factor mitogen. However, it is currently unknown how the overall anterior-to-posterior directionality of such migration is controlled. Here, we show that individual progenitor cells migrate together in a concerted, disciplined manner, a behavior that is dependent on the neighboring fat body. We identify the fat body-derived cytokine, Upd2, in targeting and inducing JAK/STAT signaling in tracheal progenitors to maintain their directional migration. Perturbation of either Upd2 production in fat body or JAK/STAT signaling in trachea causes aberrant bidirectional migration of tracheal progenitors. We show that JAK/STAT signaling promotes the expression of genes involved in planar cell polarity leading to asymmetric localization of Fat in progenitor cells. We provide evidence that Upd2 transport requires Rab5- and Rab7-mediated endocytic sorting and Lbm-dependent vesicle trafficking. Our study thus uncovers an inter-organ communication in the control of disciplined migration of tracheal progenitor cells, a process that requires vesicular trafficking of fat body-derived cytokine Upd2 and JAK/STAT signaling-mediated activation of PCP genes.

## Introduction

Adult stem cells are multipotent cell populations which inhabit their niche but mobilize to initiate tissue reconstruction during organismal growth and regeneration. An intriguing feature of stem cells is their capability of migrating in a disciplined directionality toward locations undergoing reconstruction (*Li and Clevers, 2010*). Such a highly disciplined movement is critical for maintaining tissue homeostasis and is influenced by various niche-intrinsic signals and external stimuli, and its aberrancy causes diseases such as hypertrophy (*Zhou et al., 2024*). The damaged tissue or distant organs that elicit systemic signals promote the migration of adult stem cells (*Jones and Wagers, 2008*). In addition, interactions with other cell types, soluble factors (e.g. cytokines, growth factors, and hormones) and tissue stiffness collectively bolster the mobilization of stem cells (*Fuchs and Blau, 2020*). Despite growing appreciation of adult stem cells as a primary source for tissue regeneration, the mechanism governing directional stem cell migration remains yet to be elucidated.

*Drosophila* tracheal progenitors are a population of adult stem cells that rebuild the degenerating trachea during metamorphosis. The progenitor cells reside in Tr4 and Tr5 metameres and start to move along the tracheal branch toward sites of regeneration (*Chen and Krasnow, 2014*; *Pitsouli and Perrimon, 2010*). Movement of these progenitor cells follows a stereotypical anterior-to-posterior axis (*Figure 1A*), thus representing a suitable system to investigate mechanisms controlling the directionality of stem cell migration. The activation of tracheal progenitors is stimulated by the morphogen Branchless (Bnl), fly homolog of fibroblast growth factor (FGF) (*Chen and Krasnow, 2014*), and the insulin hormone (*Li et al., 2022*). Intercellular communication and synergy between organs also contribute to the branching morphogenesis (*Perochon et al., 2021*; *Schottenfeld et al., 2010*; *Tamamouna et al., 2021*). The functional role of the interactions between trachea and other organs in modulating tracheal progenitor behavior has been largely unknown.

*Drosophila* fat body is the functional analog of mammalian adipose tissue and the major organ sensing various hormonal and nutritional signals to orchestrate systemic growth, metabolism and stem cell maintenance (*Sriskanthadevan-Pirahas et al., 2022*). Fat body produces regulatory molecules known as fat body signals (FBSs), which remotely affect the activity of other organs (*Ingaramo et al., 2020*; *Zheng et al., 2016*). For instance, the fat body-to-brain signals modulate insulin-like peptides production (*Rajan and Perrimon, 2012*), visual attention, and sleep behavior (*Ertekin et al., 2020*).

The *Drosophila* family of interleukin-6 (IL-6)-like cytokines consist of Unpaired (Upd, also called Outstretched), Upd2 and Upd3, and serve as mediators of systemic signaling. Whereas Upd1 and Upd3 derive from fly brain and plasmatocytes (*Beshel et al., 2017*; *Woodcock et al., 2015*), Upd2 is primarily produced by the fat body (*Rajan et al., 2017*), although muscle-derived Upd2 is also reported (*Zhao and Karpac, 2017*). The Upd proteins act as ligands which bind to a common GP130-like receptor, Domeless (Dome) on target cells (*Agaisse et al., 2003*; *Chen et al., 2002*). Upon association of ligands, the Dome receptors dimerize and recruit the non-receptor tyrosine kinase JAKs leading to their subsequent transactivation via phosphorylation. The transactivated JAKs then phosphorylate the tyrosine residues of their substrates, including the bound receptors and cytosolic STATs. The phosphorylation of STATs promotes their dimerization and nuclear translocation to activate transcriptional program (*Darnell, 1997*). JAK/STAT signaling requires the IL-6 cytokines (*Heinrich et al., 2003*), and is implicated in numerous cellular events including cell proliferation, differentiation, migration, and apoptosis (*O'Shea et al., 2002*).

Here, we investigate molecular basis underlying directional stem cell migration using the *Drosophila* tracheal progenitors as a model. Our results identify a cytokine-mediated inter-organ communication between fat body and the progenitor cells that is necessary for their disciplined, directional migration. The directional migration of the progenitors relies on JAK/STAT signaling and its downstream targets of planar cell polarity (PCP) components. Importantly, the Upd2 cytokines derived from fat body are transported through vesicular trafficking to induce JAK/STAT signaling in tracheal progenitors. Our study reveals that tracheal progenitors establish migratory directionality as they exit their niches and that the disciplined migration of the progenitors depends on an inter-organ signaling originating from the fat body.

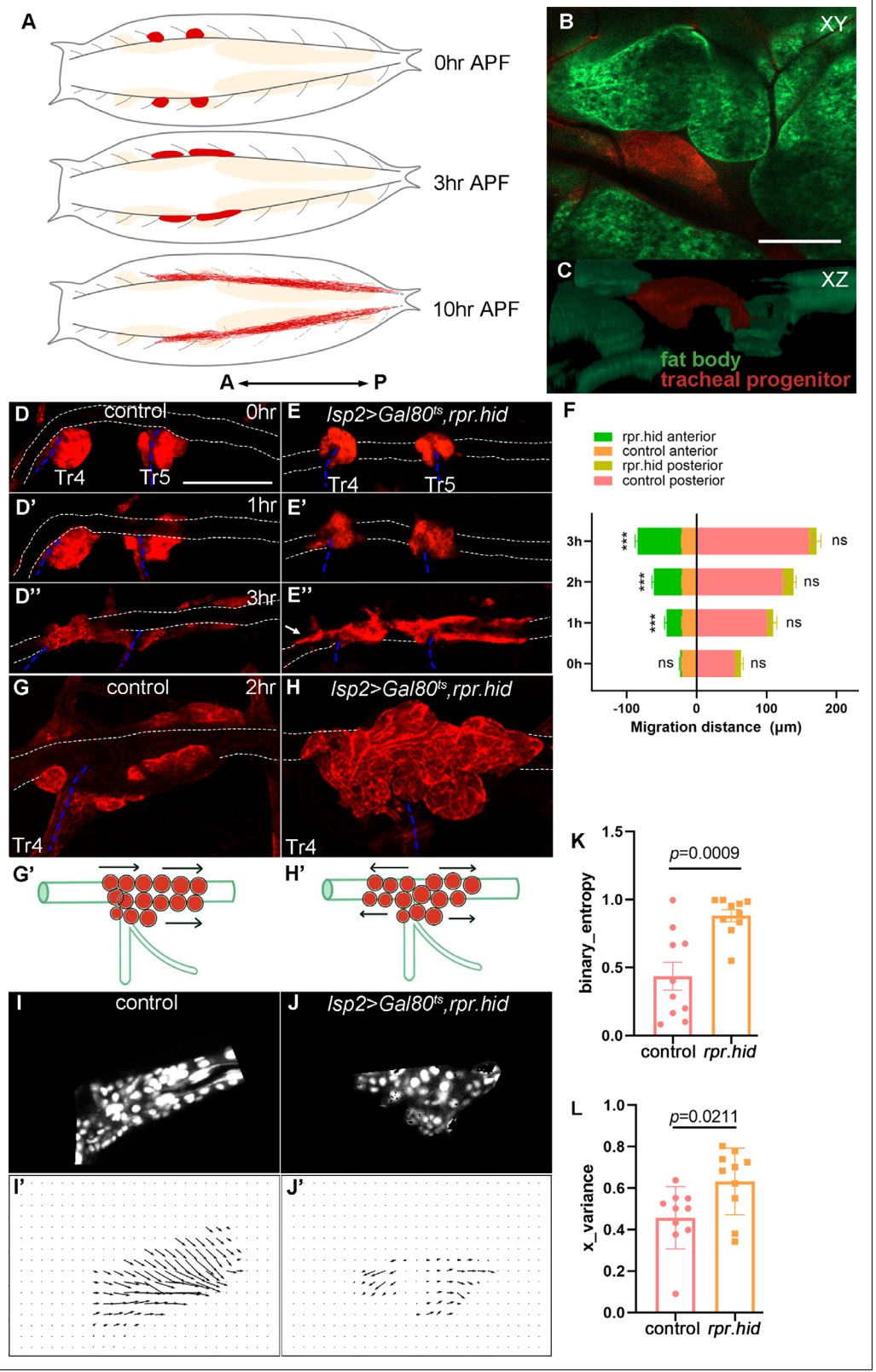

**Figure 1.** Fat body affects disciplined migration of tracheal progenitors. (**A**) Schematic cartoon showing the migration of tracheal progenitors (red) and degenerative tracheal branches (dashed gray lines) in pupae. Fat body is shown in beige. Arrows denote anterior–posterior (A–P) axis. Frontal section (**B**) and sagittal view (**C**) showing the relative position of fat body and tracheal progenitors. (**D–J′**) Migration of tracheal progenitors in control and

*Figure 1 continued on next page*

*Figure 1 continued*

fat body perturbation flies. (**D–D″**) Migration of tracheal progenitors (red) upward from transverse connective (blue dashed lines) and along the dorsal trunk (white dotted lines) at 0 hr APF (**D**), 1 hr APF (**D′**), and 3 hr APF (**D″**). Bidirectional movement of tracheal progenitors in fat body-depleted (*lsp2>rpr.hid*) flies. 0 hr APF (**E**), 1 hr APF (**E′**), and 3 hr APF (**E″**). Arrows point to anterior movement of tracheal progenitors. (**F**) Bar graph showing the migration distance of tracheal progenitors. The top chart of column represents the migration distance of anterior-most stem cells, and the lower chart of column represents the migration distance of posterior-most stem cells. Error bars represent SEM, *n* = 6. (**G, G′**) The distribution of progenitors at 2 hr APF. (**H, H″**) The distribution of progenitors in fat body-depleted flies at 2 hr APF. (**I–J′**) Computer simulation depicting trajectories of progenitor migration. (**I, J**) Confocal images of tracheal progenitors. (**I′, J′**) Vectors of progenitor migration. (**K**) Bar graph plots the binary entropy that represents the disorderedness of migration direction of tracheal progenitors. Error bars represent SEM, *n* = 10. (**L**) The Bernoulli random variable X showing optic flow distribution of the binarized directions in each group. Error bars represent SEM, *n* = 10. N.S. indicates not significant. Scale bar: 100 μm (**B, C, G**), 200 μm (**D–E″**). Genotypes: (**B, C**) *UAS-mCD8-GFP/+; lsp2-Gal4,P[B123]-RFP-moe/+*; (**D–D″, G, G′**) *Gal80^{ts}/+;lsp2-Gal4,P[B123]-RFP-moe/+*; (**E–E″, H, H′**) *UAS-rpr-hid/+;Gal80^{ts}/+;lsp2-Gal4,P[B123]-RFP-moe/+*.

The online version of this article includes the following figure supplement(s) for figure 1:

**Figure supplement 1.** Genetic perturbation of fat body by expression of pro-apoptotic genes, *rpr* and *hid*.

## Results

### Dependence of tracheal progenitors on the fat body

The fly tracheal progenitors are activated and move posteriorly along the dorsal trunk (DT) at the onset of pupariation (*Figure 1A*). We set out to delve into the underlying mechanisms of directional progenitor cell movement and tentatively surveyed organs that may coordinate this process. In *Drosophila*, the fat body resides anatomically in proximity with trachea (*Figure 1B, C*; *Video 1*) and is the principal reservoir for energy consumption. To determine whether the integrity of fat body is required for tracheal progenitors, we perturbed larval or pupal fat body by expressing pro-apoptotic cell death genes, *hid* and *reaper* (*rpr*), under the control of a fat body-specific driver, *lsp2*-Gal4 (*Cherbas et al., 2003*). Expression of *hid* and *rpr* in L3 stage impaired fat body integrity and adipocyte abundance, and generated slender larvae and pupae (*Figure 1—figure supplement 1A–D*). In these animals, the tracheal progenitors exhibited a sign of undisciplined migration and tended to move bidirectionally (*Figure 1D–F*), although their migration rate, cell number and proliferation remained unchanged (*Figure 1—figure supplement 1E–H*, *Figure 3—figure supplement 1*). The undisciplined bidirectional migration behavior of tracheal progenitors in fat body-defective animals is in stark contrast to control animals where the progenitors migrated unambiguously toward posterior (*Figure 1D–D″* and *Video 2*). To gain a quantitative view of progenitor cell migration, we traced the movement of individual cells by time-lapse confocal imaging. At 2 hr APF, tracheal progenitors from fat body deficit animals displayed a symmetrical distribution relative to the junction between DT and transverse connective (TC), compared with an

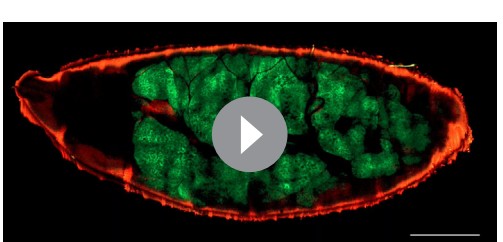

**Video 1.** 3D view of confocal image stack of tracheal progenitors and fat body. Scale bar: 100 μm. Genotype: *UAS-mCD8-GFP/+;lsp2-Gal4,P[B123]-RFP-moe/+*.
https://elifesciences.org/articles/100037/figures#video1

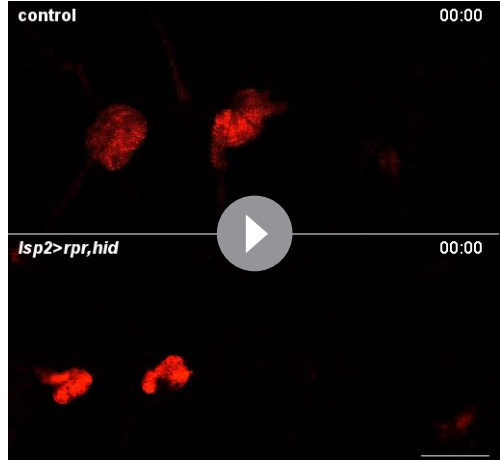

**Video 2.** The movement of tracheal progenitors in control and *rpr.hid* flies. Scale bar: 100 μm. Genotypes: *Gal80^{ts}/+;lsp2-Gal4,P[B123]-RFP-moe/+* (control) and *UAS-rpr-hid/+;Gal80^{ts}/+;lsp2-Gal4,P[B123]-RFP-moe/+*.
https://elifesciences.org/articles/100037/figures#video2

L-shape localization of niche-associated and migratory progenitors established by a posterior movement in control (*Figure 1G–H'*). Gauging the vector denoting the movement of each progenitor (*Figure 1I, I'*) revealed that the traces of progenitor groups in fat body-depleted animals exhibited a fan-shaped pattern (*Figure 1J, J'*). Owing to this undisciplined movement, entropy of the system was notably elevated upon increased inconsistency of migration vectors (*Figure 1K*). The bidirectional migratory progenitors displayed longer territory as assessed by binarized direction (*Figure 1L*). Collectively, these results suggest that fat body has an integral role in maintaining the discipline of tracheal progenitor movement.

## Upd2–JAK/STAT signaling between fat body and trachea

Since fat body impacts the behavior of tracheal progenitors, we next attempted to investigate the signal between these two interdependent organs. For this purpose, we first performed RNA sequencing (RNA-seq) analysis of tracheal progenitors from aforementioned fat body-defective flies. The results revealed a dramatical alteration of transcriptional program in tracheal progenitors upon the perturbation of fat body (*Figure 2—figure supplement 1A, B*, B). Interestingly, the functional cluster of 'cytokine activity' showed prominent enrichment in the differentially expressed genes (DEGs) in progenitors from *lsp2>rpr.hid* pupae (*Figure 2A*). This raised the possibility that certain cytokine-responsive signaling was induced in tracheal progenitors and the signaling was compromised by impairment of fat body. Therefore, we proceeded to analyze the expression of genes responsive to cytokine signaling. Analyzing the RNA-seq data revealed that the cytokine-dependent JAK/STAT and Dpp signaling were notably upregulated upon the activation of progenitors (*Figure 2B*). Importantly, fat body depletion led to suppression of target genes of JAK/STAT, PI3K, and Dpp signaling in tracheal progenitors, suggesting their dependence on the function of fat body (*Figure 2C*).

To evaluate the roles of these signaling proteins, we perturbed their expression in fat body by the expression of RNAi constructs. Knockdown of candidates including some cytokines specifically in fat body did not affect the direction of tracheal progenitor migration (*Figure 2—figure supplement 1C–J*), except for *upd2*, whose depletion phenocopied fat body ablation-induced bidirectional movement of tracheal progenitors (*Figure 2D–F* and *Video 3*). These results suggest a role of fat body-produced Upd2 in remotely regulating the tracheal progenitors.

Then, we performed surface proteome in vivo (*Li et al., 2020*) to investigate the spectrum of molecules received by trachea (*Figure 2—figure supplement 2*). The trachea-associated proteins were biotinylated through a reaction mediated by a membrane-tethered horse radish peroxidase (HRP-CD2) (*Figure 2—figure supplement 2A–D*). Of the 1684 streptavidin-precipitated proteins captured by mass spectrometry (*Figure 2G*), a functional cluster enriched for receptor signaling via JAK/STAT was identified (*Figure 2H*). The JAK/STAT pathway is one of the principal cellular signaling that responds to Upd2 ligand (*Hombría et al., 2005*). *Drosophila* JAK/STAT signaling is well conserved (*Arbouzova and Zeidler, 2006*; *Zeidler et al., 2000*) and comprises a single JAK (Hopscotch, Hop) and one STAT (Stat92E), in contrast to a handful of homologues (four JAK and seven STAT genes) found in mammals. Domeless (Dome), the receptor for JAK/STAT pathway, exhibited pronounced expression in the tracheal progenitors (*Figure 2—figure supplement 2E*). To test if these Dome receptors actively interact with their ligands, we adapted a technique to monitor ligand–receptor interaction in vivo (*Michel et al., 2011*) and constructed a Dome variant (DIPF) which only fluoresces in the ligand-binding and phosphorylated state (*Figure 2I*). The signal of this DIPF reporter was detected in both larval fat body and salivary gland (*Figure 2—figure supplement 2F, H*), which is consistent with active JAK signaling implicated in the development of the tissues (*Chakrabarti et al., 2016*; *Krautz et al., 2020*). When expressed in the tracheal system, DIPF displayed robust fluorescent signal in the tracheal progenitors (*Figure 2J*). These data suggest that receptor signaling of JAK/STAT is active in the tracheal progenitors.

To analyze the functional importance of JAK/STAT signaling in tracheal progenitors, we perturbed the principal components of this signaling, namely the receptor Dome, signal transducer Hop, or the downstream transcription factor Stat92E, by *btl*-Gal4-driven expression of RNAi constructs. Under these conditions in which JAK/STAT pathway is compromised, the tracheal progenitors aberrantly migrated anteriorly, which is reminiscent of *upd2* loss-of-function in the fat body (*Figure 3A–E*, *Video 4*, and *Figure 3—figure supplement 1A–D*). The aberrant anterior migration of tracheal progenitors upon perturbation of JAK/STAT components led to incomplete regeneration of airway and impairment of

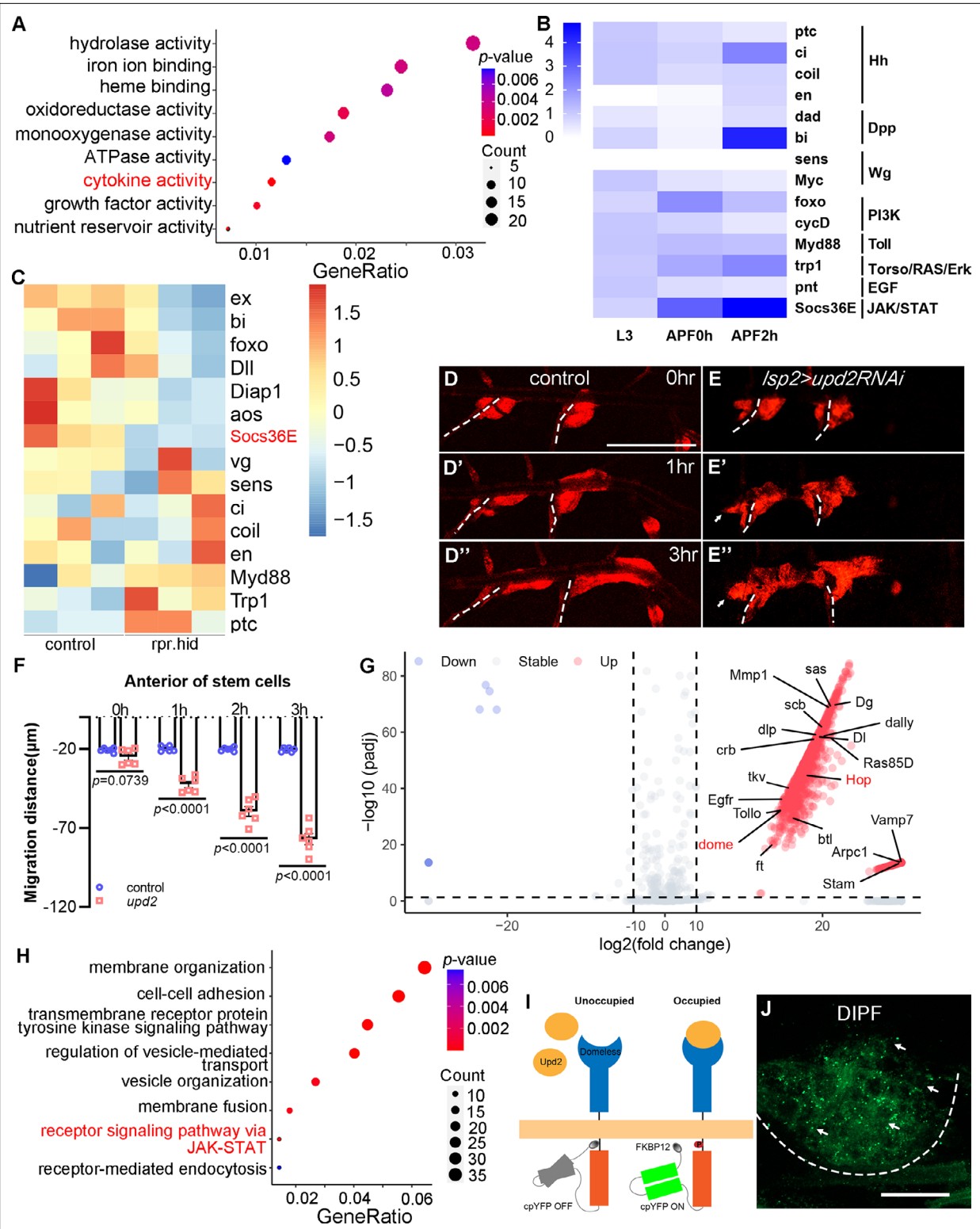

**Figure 2.** Dependence of tracheal progenitors on cytokines from fat body. (**A**) Top functional clusters among the differentially expressed genes of progenitors between control and fat body-depleted pupae. Gene ratio refers to the proportion of genes in a dataset that are associated with a particular biological process, function, or pathway. Count indicates the number of genes from an input gene list that are associated with a specific GO term. (**B**) Heatmap depicting expression levels of principal target genes of signaling pathways in L3 larvae, 0 hr APF pupae and 2 hr APF pupae. (**C**) Heatmap showing the differential expression of target genes of signaling pathways between control and fat body-depleted pupae. Migration of tracheal progenitors along the dorsal trunk at 0 hr APF (**D**), 1 hr APF (**D'**), and 3 hr APF (**D''**). The white dashed line shows transverse connective.

*Figure 2 continued on next page*

*Figure 2 continued*

(**E–E″**). Migration of tracheal progenitors in *upd2RNAi* flies. (**F**) Bar graph plots the migration distance of tracheal progenitors. Error bars represent SEM, *n* = 6. (**G**) Volcano plot showing surface proteomics of tracheal epithelium (upregulated genes with tenfold or higher changes in red; downregulated genes with tenfold or higher changes in blue). (**H**) Top functional classes among the surface proteomics of trachea. (**I**) Schematic diagram depicting the working principle of the DIPF reporter. (**J**) The signal of DIPF reporter in tracheal progenitors. The progenitors are outlined by dashed lines. N.S. indicates not significant. Scale bar: 200 μm (**D–E″**), 50 μm (**J**). Genotypes: (**A, C**) *lsp2-Gal4,P[B123]-RFP-moe/+* for control, *UAS-rpr-hid/+;Gal80ts/+;lsp2-Gal4,P[B123]-RFP-moe/+*; (**B**) *P[B123]-RFP-moe/+*. (**D–D″**) *lsp2-Gal4,P[B123]-RFP-moe/+*; (**E–E″**) *lsp2-Gal4,P[B123]-RFP-moe/UAS-upd2RNAi*; (**J**) *btl-Gal4/UAS-DIPF*.

The online version of this article includes the following source data and figure supplement(s) for figure 2:

**Figure supplement 1.** Identification of genes that contribute to progenitors migration.

**Figure supplement 2.** Surface proteomics of *Drosophila* trachea.

**Figure supplement 2—source data 1.** Original files for western blot analysis displayed in *Figure 2—figure supplement 2D*.

**Figure supplement 2—source data 2.** PDF file containing original western blots for *Figure 2—figure supplement 2D*, indicating the relevant bands and treatments.

tracheal integrity and caused melanization in the trachea (*Figure 3—figure supplement 1E–I*). In agreement with genetic perturbation of JAK/STAT signaling, pharmacological inhibition of JAK by a small-molecule inhibitor, Tofacitinib (*Palmroth et al., 2021*), also triggered bidirectional movement of tracheal progenitors (*Figure 3F–H*). The bidirectional movement was not due to excessive progenitors or crowding (*Figure 3—figure supplement 2A*).

Concurrently, the activity of JAK/STAT pathway, as assessed by the Stat92E-GFP reporter (*Bach et al., 2007*), was substantially impaired when components of the pathway were depleted (*Figure 3—figure supplement 2B–E, J*). To determine whether the tracheal JAK/STAT signaling depends on fat body-derived Upd2, we depleted Upd2 in fat body and observed that Stat92E-GFP signal in tracheal progenitors was severely decreased, suggesting that JAK/STAT signaling in the trachea requires fat body-produced Upd2 (*Figure 3—figure supplement 2F, G, J*). Consistently, inhibition of JAK/STAT signaling using Tofacitinib reduced the expression of Stat92E-GFP (*Figure 3—figure supplement 2H–J*). Taken together, these observations suggest that Upd2-responsive JAK/STAT signaling in the trachea is essential for the disciplined migration of progenitors.

## Genes regulated by JAK/STAT signaling in the trachea

To gain a comprehensive understanding of the molecular details underlying the discipline of tracheal progenitor migration, we conducted genomic chromatin immunoprecipitation (ChIP-seq) to identify loci bound by Stat92E which functions as the transcription factor of JAK/STAT pathway. This revealed a total of 21,312 Stat92E binding peaks, ~95.7% of which located within 2 kb of transcription start sites of annotated genes (*Figure 4—figure supplement 1A*). In particular, 86% of the peaks (18,328 peaks) were enriched either in promoter regions or within gene bodies, and 66.1% of the peaks (13,490 peaks) resided near the 5′ ends of annotated genes, namely in the promoter regions, first exons and first introns (*Figure 4—figure supplement 1B, C*). GO analysis of putative target genes of Stat92E identified one cluster associated with establishment of planar polarity (*Figure 4A*). In line with this, the functional class associated with establishment of planar polarity was also abundantly represented among the DEGs upon the activation of tracheal progenitors in larval–pupal transition (*Figure 4B*). Notably, Stat92E binding was detected in the promoters and intronic regions

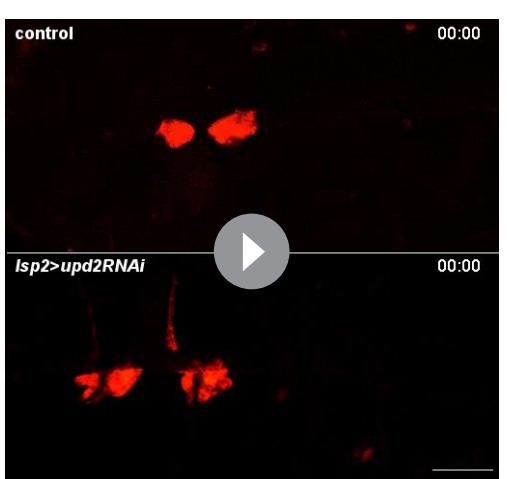

**Video 3.** The movement of tracheal progenitors in control and *upd2RNAi* flies. Scale bar: 100 μm. Genotypes: *lsp2-Gal4,P[B123]-RFP-moe/+* (control) and *lsp2-Gal4,P[B123]-RFP-moe/UAS-upd2RNAi*.
https://elifesciences.org/articles/100037/figures#video3

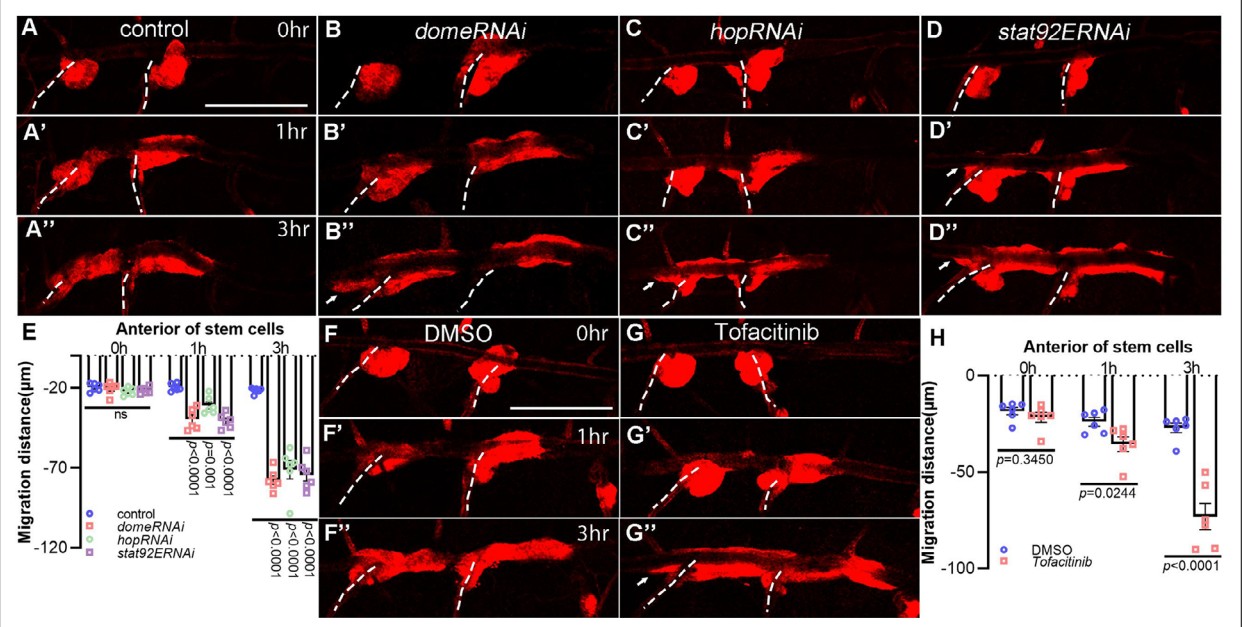

**Figure 3.** JAK/STAT pathway is required for the discipline of tracheal progenitor migration. (**A–D″**) Migration of tracheal progenitors along the dorsal trunk at 0 hr APF, 1 hr APF, and 3 hr APF. The white dashed line shows transverse connective. The progenitors of control (**A–A″**), *domeRNAi* (**B–B″**), *hopRNAi* (**C–C″**), and *stat92ERNAi* (**D–D″**) flies. (**E**) Bar graph showing migration distance of progenitors. Error bars represent SEM, n = 6. (**F–G″**) JAK inhibition causes bidirectional movement of progenitors. Migration of tracheal progenitors in the absence (DMSO-fed) (**F–F″**) or in the presence of Tofacinib (JAK inhibitor) (**G–G″**). (**H**) Bar graph showing the distance of anterior movement. Error bars represent SEM, n = 6. Scale bar: 200 μm (**A–D″, F–G″**). Genotypes: (**A–A″, F–G″**) btl-Gal4/+;P[B123]-RFP-moe/+; (**B–B″**) btl-Gal4/+;P[B123]-RFP-moe/UAS-domeRNAi; (**C–C″**) btl-Gal4/+;P[B123]-RFP-moe/UAS-hopRNAi; (**D–D″**) btl-Gal4/+;P[B123]-RFP-moe/UAS-stat92ERNAi.

The online version of this article includes the following figure supplement(s) for figure 3:

**Figure supplement 1.** The roles of JAK/STAT pathway in tracheal development.

**Figure supplement 2.** The dependence of tracheal progenitors on JAK/STAT pathway.

of genes functioning in distal-to-proximal signaling (*Cho and Irvine, 2004*), such as *dachsous* (*ds*), *four-jointed* (*fj*), *fz*, *stan*, *Vang*, and *fat2* (*Figure 4C*). Additionally, Stat92E occupied in the promoter regions of *crb* and *yurt,* two genes involved in apical–basal polarity and tracheal tube growth (*Laprise et al., 2006*; *Schottenfeld-Roames and Ghabrial, 2012*; *Schottenfeld-Roames et al., 2014*; *Figure 4—figure supplement 1D*). The enrichment of Stat92E in the promoters and/ or regulatory regions of these putative targets was confirmed by ChIP-qPCR (*Figure 4—figure supplement 1E*).

To further validate these putative Stat92E targets and investigate their dependence on JAK/STAT signaling, we analyzed their expression from several fosmid transgenes which have a GFP tag fused to *ds*, *fj*, or *ft* and express at endogenous levels. Ds and Fj were abundant in the progenitor cells, but were vastly reduced upon depletion of *dome*, *hop*, or *stat92E*, suggesting that they are regulated by JAK/STAT pathway (*Figure 4D–M*). Furthermore, it is reported that the function of Ft is influenced by cell-autonomous increase of Ds level and its protein level is enhanced by Ds reduction (*Ambegaonkar et al., 2012*; *Matakatsu and Blair, 2004*), which is also evidenced by our analysis using *dsRNAi* and *UAS-ds* in the tracheal progenitors (*Figure 4—figure supplement 2A–D*). In accordance with this notion, the level of Ft, as assayed by the Ft-GFP reporter, was elevated by the reduction of JAK/ STAT signaling (*Figure 4N–R*). We also analyzed GFP-tagged fosmid transgenes of *fat2*, *crb*, and *yurt* and found that they were discernably reduced upon impairment of JAK/STAT signaling, suggesting that they are also regulated by JAK/STAT pathway (*Figure 4—figure supplement 1F–T*). Additionally, the transcription of *ds*, *fj*, *ft*, *fat2*, *crb*, and *yurt* was compromised by expression of *stat92ERNAi* (*Figure 4—figure supplement 1U*). In sum, these results suggest that JAK/STAT promotes components involved in the establishment of polarity in tracheal cells.

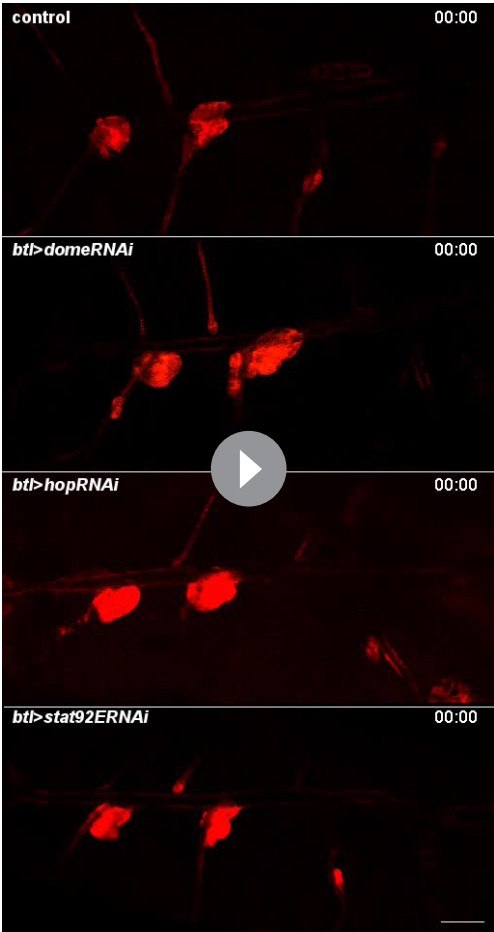

**Video 4.** The movement of tracheal progenitors in control and JAK/STAT pathway-perturbed flies. Scale bar: 100 μm. Genotypes: *btl-Gal4/+;P[B123]-RFP-moe/+* (control), *btl-Gal4/+;P[B123]-RFP-moe/UAS-domeRNAi*, *btl-Gal4/+;P[B123]-RFP-moe/UAS-hopRNAi*, and *btl-Gal4/+;P[B123]-RFP-moe/UAS-stat92ERNAi*.
https://elifesciences.org/articles/100037/figures#video4

## The roles of JAK/STAT targets in the disciplined migration

To evaluate the functional roles of the polarity proteins in tracheal progenitor migration, we perturbed their expression in the tracheal progenitors by expressing RNAi against *ds*, *ft*, or *fj*, which were identified by ChIP-seq as the targets of JAK/STAT. In these flies, tracheal progenitors exhibited bidirectional movement, which is reminiscent of the impairment of JAK/STAT signaling (*Figure 5A–E* and *Video 5*). Similar observations were obtained by over-expression of *ft* or *ds* in the trachea (*Figure 5—figure supplement 1A–D*), consistent with previous reports that both loss- and gain-of-function of PCP components disrupt the PCP (*Adler et al., 2000*; *Tree et al., 2002*; *Vinson and Adler, 1987*). The disciplined migration of tracheal progenitors was also impaired by the expression of *fat2RNAi*, *crbRNAi*, *yurtRNAi*, or *scbRNAi* (*Figure 5—figure supplement 1E–I*), but was not affected by perturbation of molecules involved in cell adhesion such as Enabled (Ena), Fak, E-cadherin, and Robo2 (*Figure 5—figure supplement 1J*).

Migratory cells generate protrusions at the leading edge to initiate movement (*Cetera et al., 2014*). The normal posteriorly migrating tracheal progenitors extend protrusions toward the migratory directions (*Figure 5—figure supplement 2A*), but in the bidirectionally moving progenitors in which *upd2* in fat body was perturbed, extensive filopodia were projected from both the anterior and posterior fronts (*Figure 5—figure supplement 2B, C*), indicating that the aberrantly anteriorly moving progenitors may adopt the identity as those moving posteriorly. Bidirectionally migrating progenitors induced by perturbation of JAK/STAT signaling did not alter the expression of the tracheal inducer, *branchless*

(*bnl*) (*Figure 5—figure supplement 2D, E*). Further analysis revealed that the progenitors exhibited elevated levels of Ft at the leading edge where they attached to DT (*Figure 5F, G*). Accordingly, progenitors that underwent bidirectional movement exhibited pronounced abundance of Ft at both the anterior and posterior frontal edges (*Figure 5H–J*). To further evaluate the functional roles of Ft–Ds–Fj module in disciplined migration, we utilized the high-mobility carcinoma cells, SKOV-3, and found that perturbation of Fj that phosphorylates the extracellular cadherin domains of both Ft and Ds and modifies their heterophilic binding (*Thomas and Strutt, 2012*), Ft or Ds concurrently displayed compromised directionality and reduced consistency of movement in a two-dimensional culture (*Figure 5K–M*, *Figure 5—figure supplement 3*). Together with the results in previous sections, these observations suggest that the activated tracheal progenitors establish a disciplined migration through the asymmetrical distribution of polarity proteins which is directed by an Upd2–JAK/STAT signaling stemming from the remote organ of fat body.

## Upd2 in the fat body-produced vesicle

Besides the JAK/STAT signaling, another functional class enriched for vesicle-mediated transport was prominent from our surface proteome analysis of the trachea (*Figure 2H*). A series of components

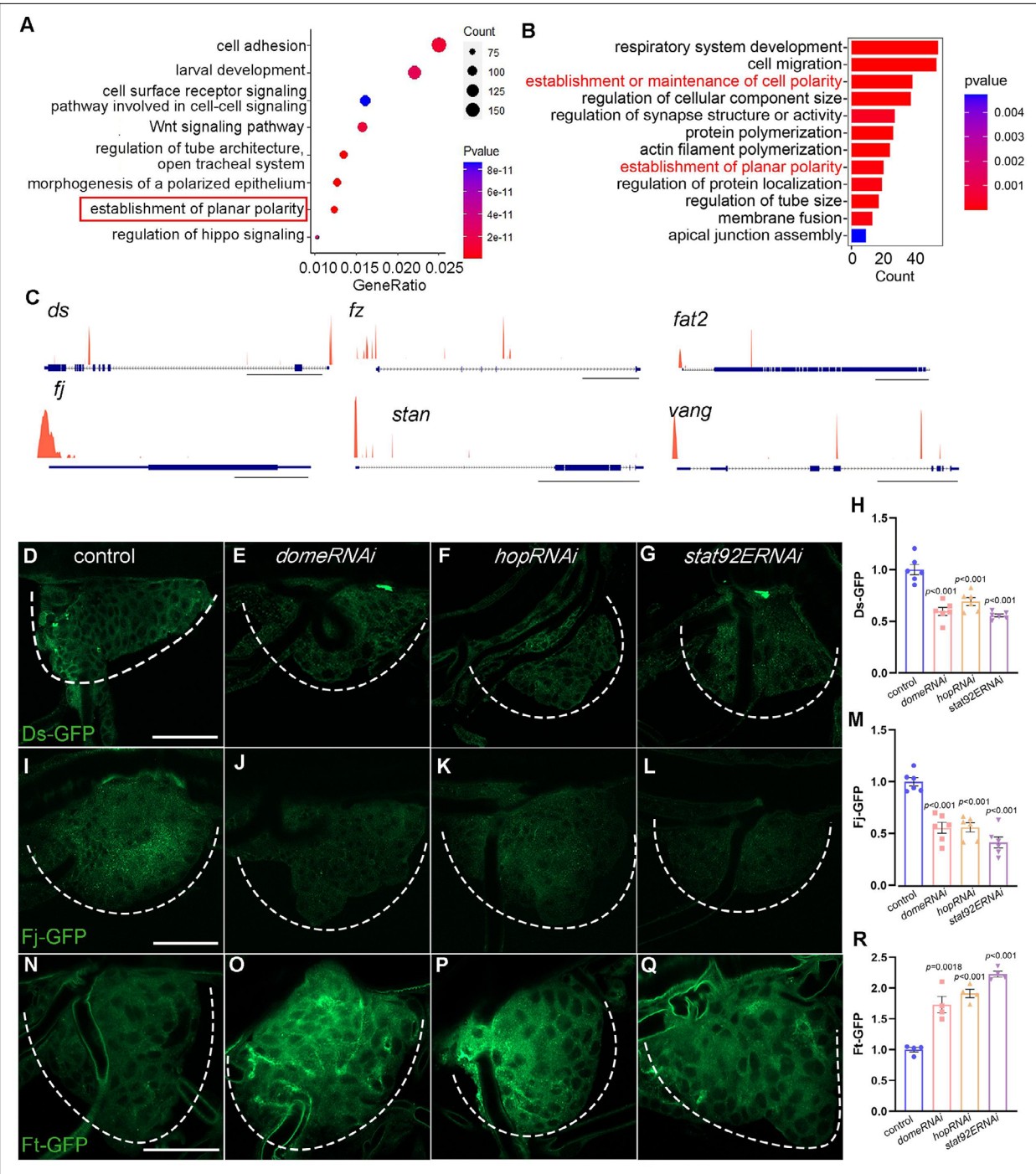

**Figure 4.** Identification of gene targets of Stat92E in *Drosophila* tracheal progenitors. (**A**) Bubble plot represents the top functional clusters among gene targets. The establishment of planar polarity denoted in red solid box is identified with high enrichment score. (**B**) Top functional classes among the differentially expressed genes in larval–pupal transition. (**C**) ChIP-seq peaks at loci regulated by Stat92E. Scale bar: 20 kb (*ds*, *fz*, *stan*), 5 kb (*fat2*, *vg*), 1 kb (*fj*). (**D–H**) Validation of gene targets of Stat92E ChIP-seq. The expression of Ds-GFP in the tracheal progenitors of control (**D**), *domeRNAi* (**E**), *hopRNAi* (**F**), and *stat92ERNAi* (**G**). The progenitors are outlined by dashed lines. (**H**) The bar graphs plot the relative level of Ds. Error bars represent SEM, *n* = 6. (**I–M**) The expression of Fj in tracheal progenitors. The expression of Fj-GFP in the tracheal progenitors of control (**I**), *domeRNAi* (**J**), *hopRNAi* (**K**), and *stat92ERNAi* (**L**). Dashed lines outline tracheal progenitors. (**M**) The bar graphs plot the relative level of Fj. Error bars represent SEM, *n* = 6. (**N–R**) The level of Ft-GFP in the tracheal progenitors of control (**N**), *domeRNAi* (**O**), *hopRNAi* (**P**), and *stat92ERNAi* (**Q**). (**R**) The bar graphs plot the relative level of Ft. Error bars represent SEM, *n* = 4. Scale bar: 50 μm (**D–G, I–L, N–Q**). Genotypes: (**D**) *btl-Gal4,Ds-GFP/+*; (**E**) *btl-Gal4,Ds-GFP/+;UAS-domeRNAi/+*; (**F**) *btl-Gal4,Ds-GFP/+;UAS-hopRNAi/+*; (**G**) *btl-Gal4,Ds-GFP/+;UAS-stat92ERNAi/+*; (**I**) *btl-Gal4,Fj-GFP/+*; (**J**) *btl-Gal4,Fj-GFP/+;UAS-domeRNAi/+*; (**K**)

*Figure 4 continued on next page*

*Figure 4 continued*

btl-Gal4,Fj-GFP/+;UAS-hopRNAi/+; (**L**) btl-Gal4,Fj-GFP/+;UAS-stat92ERNAi/+; (**N**) btl-Gal4/+;Ft-GFP/+; (**O**) btl-Gal4/+;Ft-GFP/UAS-domeRNAi; (**P**) btl-Gal4/+;Ft-GFP/UAS-hopRNAi; (**Q**) btl-Gal4/+;Ft-GFP/UAS-stat92ERNAi.

The online version of this article includes the following figure supplement(s) for figure 4:

**Figure supplement 1.** Annotation and analysis of genomic occupancy of Stat92E.

**Figure supplement 2.** Ds alters the level of Ft.

that function in vesicle trafficking were identified. It has been reported that IL-6 cytokines tend to be encapsulated in secretory vesicles (*Kandere-Grzybowska et al., 2003*; *Verboogen et al., 2018*). To visualize Upd2 production and investigate its transportation kinetics, an upd2-mCherry transgene was developed and expressed under the control of *lsp2*-Gal4, which enabled tracking the dynamics of Upd2 in fat body (*Figure 6A*). In agreement with Upd2 being transported through vesicles, administration of L3 larvae with Brefeldin A (BFA), which pharmacologically inhibits vesicle formation and transport, sequestered Upd2 proteins in fat body (*Figure 6B, C*). To track the destination of the Upd2-containing vesicles, we examined mCherry signals in adjacent tissues and detected considerable amount of Upd2 puncta in the tracheal progenitors (*Figure 6D*). BFA treatment reduced Upd2-mCherry puncta in the tracheal progenitors, suggesting that tracheal progenitors receive vesicular Upd2 from the fat body (*Figure 6E, F*). Perturbation of Grasp65, a Golgi reassembly stacking protein previously implicated in Upd2 secretion (*Rajan et al., 2017*), also led to sequestration of Upd2-containing vesicles in fat body (*Figure 6G, H, J*). The vesicle formation, function, and extracellular movement are dependent on the tetraspanin superfamily proteins (*Andreu and Yáñez-Mó, 2014*). We surveyed all the tetraspanin orthologs in fly for potential roles in Upd2 vesicle formation and transport. When expressing *lbmRNAi* in fat body, Upd2-containing vesicles were vastly increased (*Figure 6I, J*). Meanwhile, perturbation of vesicle secretion or transport by expressing *grasp65RNAi* or *lbmRNAi* in fat body eliminated the presence of fat body-origin Upd2 in the trachea, suggesting that fat body-produced Upd2-containing vesicles function cell non-autonomously and contribute to other tissues/organs (*Figure 6K–N*). It should be noted that knockdown of *upd2* in the trachea did not alter the discipline of tracheal progenitor migration (*Figure 2—figure supplement 1K–M*). Collectively, these results suggest that fat body-produced Upd2 undergoes vesicle-mediated trafficking.

## The vesicular transport in JAK/STAT signaling

The results in previous section suggest that the ligand of JAK/STAT signaling is transported in a manner that depends on vesicle trafficking. To validate the role of fat body-produced vesicles in inter-organ signaling, we used genetic and pharmacological tools to perturb different processes of vesicle trafficking in fat body and monitored JAK/STAT signaling in the tracheal progenitors. Expression of *grasp65RNAi* in fat body reduced the activity of JAK/STAT signaling in the trachea, as assessed by the Stat92E-GFP reporter (*Figure 7A, B*). Similarly, RNAi targeting expression of *lbm* in fat body vanished JAK/STAT signal transduction in the trachea (*Figure 7C*). Rab GTPases coordinate vesicle trafficking and production (*Stenmark, 2009*) and have been shown to play pivotal roles in the regulation of intracellular trafficking of FGFR and EGFR (*Letizia et al., 2023*; *Olivares-Castiñeira and Llimargas, 2017*), and were identified in the surface proteome analysis. Consistently, knockdown of *rab5* or *rab7* in fat body reduced the activity of JAK/STAT signaling in the progenitor cells (*Figure 7D–F*). Corroborating the genetic manipulations, BFA treatment that impeded vesicular transport also resulted in impairment of JAK/STAT signaling in trachea (*Figure 7G–I*). Taking advantage of the aforementioned DIPF reporter to assess the response of receiving cells to ligands, we found that the fluorescent signal of DIPF was compromised upon the presence of BFA, but was unaffected by inhibitors that target the downstream JAK protein (*Figure 7J–M*), suggesting that signaling ligands are less abundant in the recipient progenitor cells and that the vesicle-mediated transport of ligands is essential for JAK/STAT signaling.

Phenotypically, the tracheal progenitors exhibited bidirectional migration in BFA-treated flies, which phenocopies JAK/STAT loss-of-function (*Figure 7N–P*). In concord with this observation, depletion of *grasp65* or *lbm* also led to bidirectional movement (*Figure 7Q–S"*, and *Video 6*). Similar observations were made in tracheal progenitors with either *rab5* or *rab7* knockdown (*Figure 7T–V*), whereas perturbation of neither *rab2* nor *rab3* affected the disciplined progenitor migration (*Figure 7—figure*

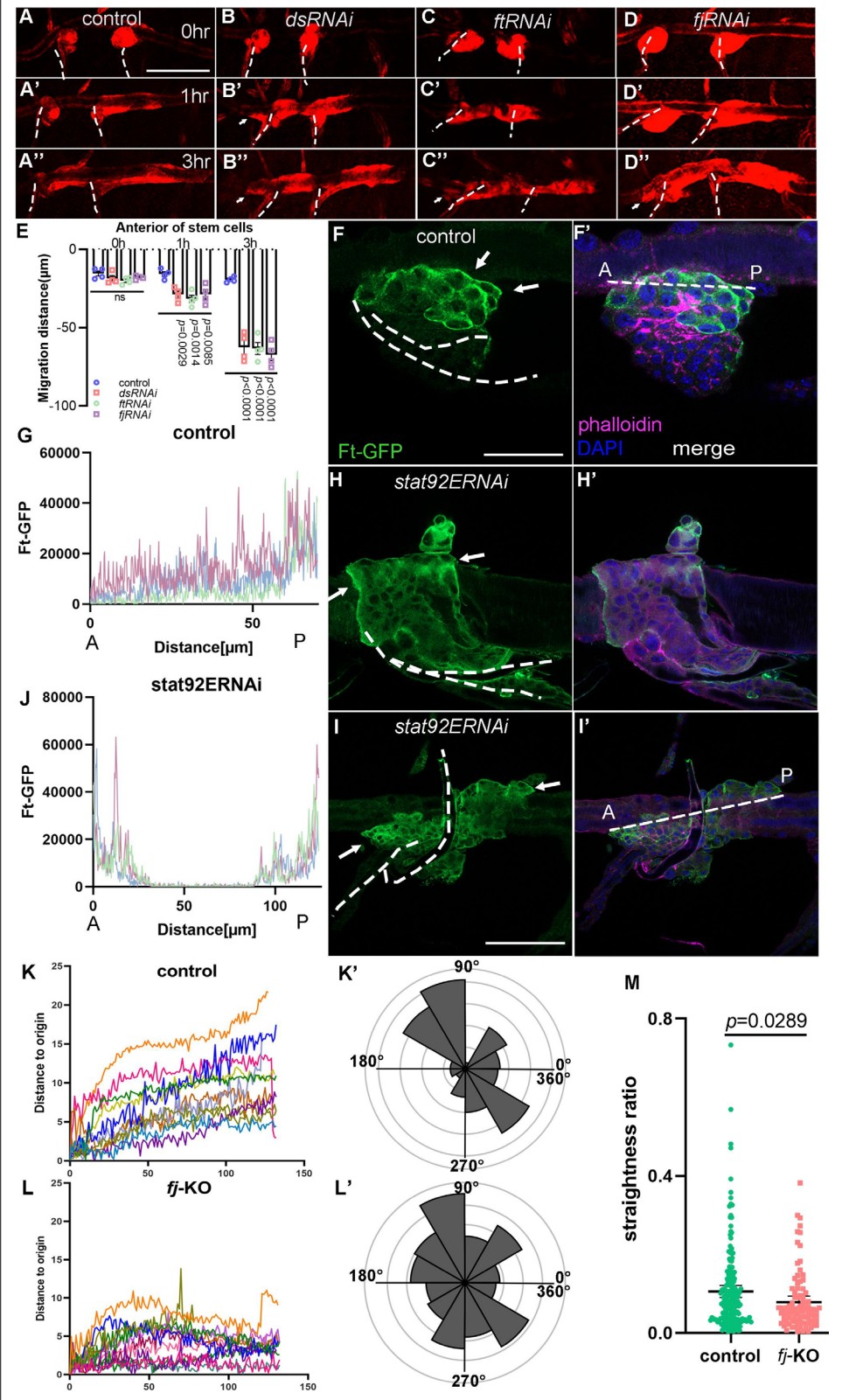

**Figure 5.** Disciplined migration requires planar cell polarity system. (**A–E**) Migration of tracheal progenitors. The migration of progenitors in control (**A–A''**), *dsRNAi* (**B–B''**), *ftRNAi* (**C–C''**), and *fjRNAi* (**D–D''**) flies. (**E**) Bar graph plots the migration distance of anterior movement. Error bars represent SEM, *n* = 4. Level of Ft in tracheal progenitors of control (**F, G**) and *stat92ERNAi* (**H–J**) flies. The images show progenitors at 1 hr APF (**H, H'**) and 2 hr

*Figure 5 continued on next page*

*Figure 5 continued*

APF (**I, I'**). Ft-GFP (green) (**F, H, I**), phalloidin (magenta), Hoechst (blue), and merged images (**F', H', I'**). Profile plots showing the level of Ft-GFP in control (**G**) and *stat92ERNAi* (**J**) flies, n = 5. ANOVA test: p < 0.0001. The levels of Ft were measured along the dotted lines in F' or I'. Anterior (A) and posterior (P). (**K**) Representative traces plot the migration distance relative to the origin, n = 12. The x-axis represents the number of captured images. Individual frame is captured every 5 min. (**K'**) Rose plot depicting the direction of cell movement. (**L**) Representative traces showing the movement of individual *fj*-KO cells relative to their origin, n = 11. The x-axis represents the number of captured images. Individual frame is captured every 5 min. (**L'**) Rose plot depicting the movement direction of *fj*-KO cells. (**M**) Scatter plots represent the ratio (*d/D*) of straight-line length displacement (*d*) relative to the length of the migration track (*D*) of individual cell. Error bars represent SEM. N.S. indicates not significant. Scale bar: 200 μm (**A–D"**), 50 μm (**F, F'**), 100 μm (**H–I'**). Genotypes: (**A–A"**) *btl-Gal4/+;P[B123]-RFP-moe/+*; (**B–B"**) *btl-Gal4/UAS-dsRNAi;P[B123]-RFP-moe/+*; (**C–C"**) *btl-Gal4/+;P[B123]-RFP-moe/UAS-ftRNAi*; (**D–D"**) *btl-Gal4/UAS-fjRNAi;P[B123]-RFP-moe/+*; (**F, F'**) *btl-Gal4/+;Ft-GFP/+*; (**H, H', I, I'**) *btl-Gal4/+;Ft-GFP/UAS-stat92ERNAi*.

The online version of this article includes the following figure supplement(s) for figure 5:

**Figure supplement 1.** Apico-basal polarity proteins contribute to the migration polarity of progenitors.

**Figure supplement 2.** Protrusions extending from the leading edge of progenitors.

**Figure supplement 3.** The dependence of migration directionality on Ft/Ds system.

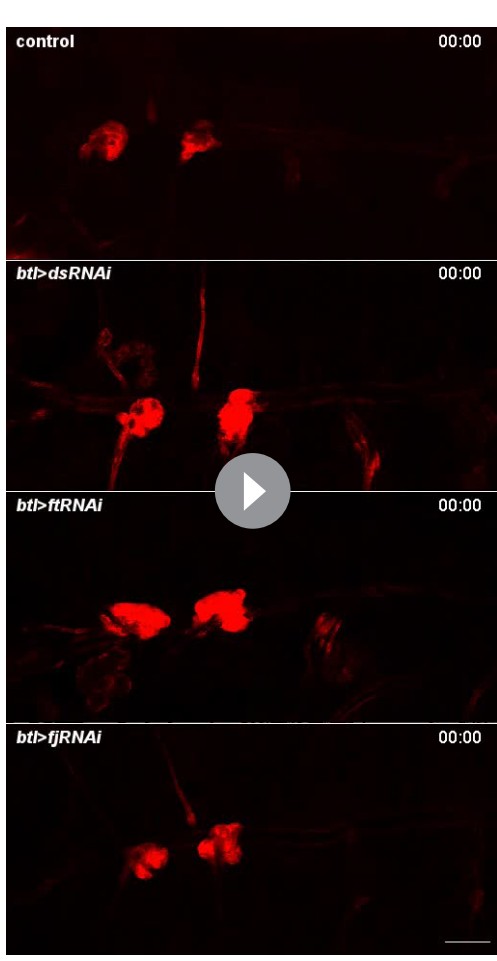

**Video 5.** The movement of tracheal progenitors in control and planar cell polarity (PCP) component aberrant flies. Scale bar: 100 μm. Genotypes: *btl-Gal4/+;P[B123]-RFP-moe/+* (control), *btl-Gal4/UAS-dsRNAi;P[B123]-RFP-moe/+*, and *btl-Gal4/+;P[B123]-RFP-moe/UAS-ftRNAi*.

https://elifesciences.org/articles/100037/figures#video5

*supplement 1A–C*). Taken together, these results suggest that JAK/STAT signaling in the trachea is dependent on the vesicle-mediated transport of its ligands from fat body.

## The interaction between Upd2 and endocytic machinery

Our results described thus far suggest that Upd2 emanating from fat body signals to JAK/STAT signaling in the trachea. To further explore the molecular basis underlying the vesicular transport of Upd2, we monitored Rab5-GFP and Rab7-GFP in fat body, which mark early and late endosomes, respectively (*Vonderheit and Helenius, 2005*). The fat body-produced Upd2 appeared vesicular (*Figure 8A*) and both Rab5 and Rab7 were found adjacent to the Upd2-harboring vesicles, suggesting that both Rab GTPases function in the transport of Upd2 (*Figure 8A'–B"*). In contrast, Rab3 exhibited non-overlapping distribution with Upd2 (*Figure 7—figure supplement 1E–E"*). Furthermore, we observed that Grasp65 was in close proximity to Upd2-containing vesicles, indicating its integral roles in these vesicles (*Figure 8C–C"*). In addition, Upd2 was observed to colocalize with the tetraspanin, Lbm (*Figure 8D–D"*). At higher resolution, Upd2 and Lbm showed close association in a supramolecular configuration (*Figure 8D''', D''''*), corroborating its role in the transport of Upd2. To determine if Upd2 interacts with the coordinators of vesicle trafficking, we employed the Duolink in situ proximity ligation assay (PLA) which revealed strong interactions between Upd2 and Rabs, such as Rab5 and Rab7 (*Figure 8E–H*), as well as Lbm (*Figure 8I, J*). The interaction was further validated by the revelation that Upd2 co-immunoprecipitated with

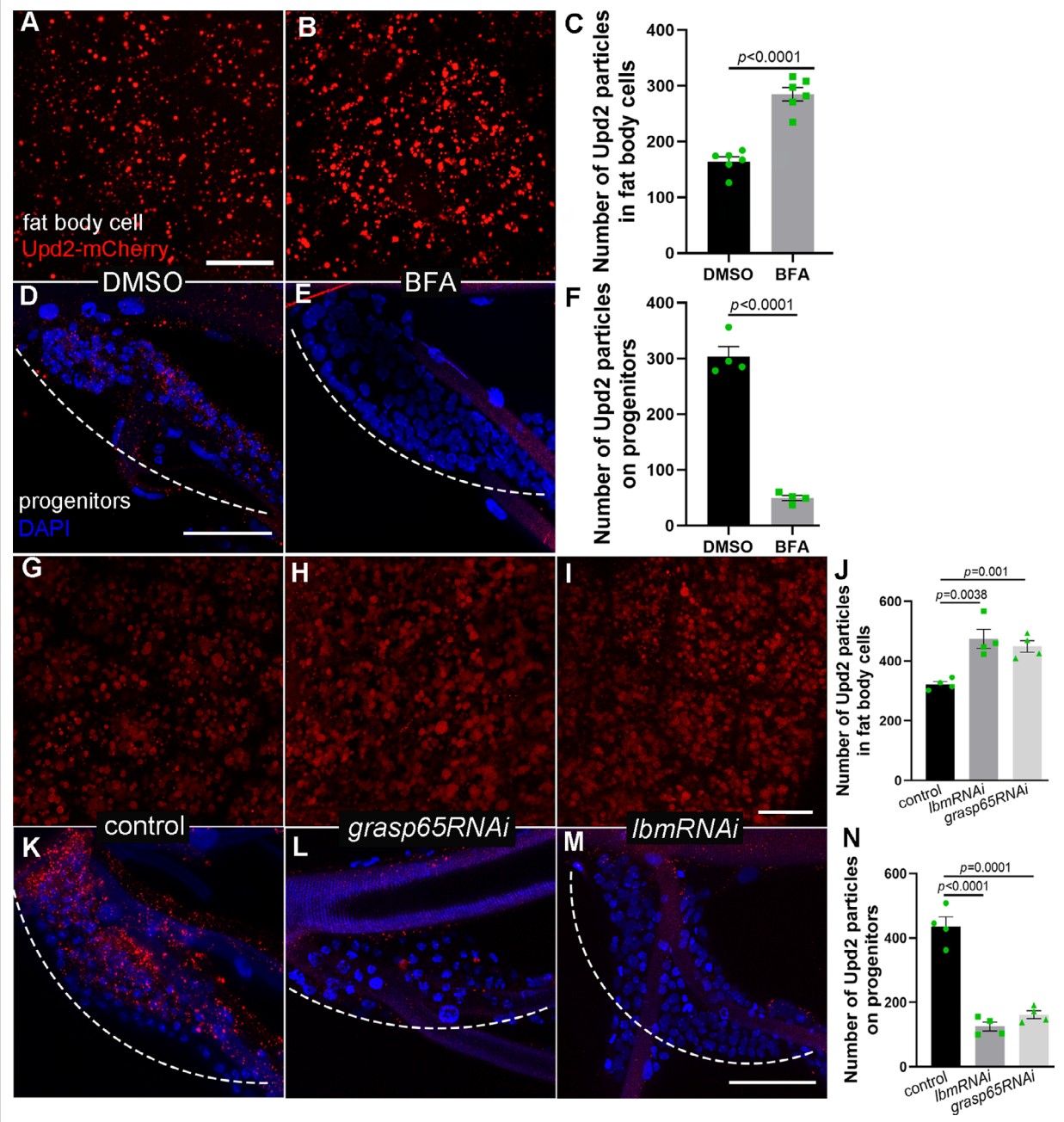

**Figure 6.** The production and transport of Upd2 from fat body. The number of Upd2-mCherry-containing vesicles in fat body of control DMSO-fed (**A**) and BFA-treated L3 larvae (**B**). Larger view in lower magnification is provided in *Figure 6—figure supplement 1*. (**C**) Bar graph plots the number of Upd2-mCherry-containing vesicles in fat body. Error bars represent SEM, *n* = 6. The confocal image showing the number of Upd2-containing vesicles in progenitors of DMSO-fed control (**D**) and BFA-treated flies (**E**). (**F**) Bar graph plots the number of Upd2-mCherry-containing vesicles in progenitors. Error bars represent SEM, *n* = 4. (**G**) The number of Upd2-mCherry containing vesicles (red) in fat body. Upd2 accumulation in fat body was increased in the presence of *grasp65RNAi* (**H**) and *lbmRNAi* (**I**). (**J**) Bar graph plots the number of Upd2-mCherry-containing vesicles. Error bars represent SEM, *n* = 4. The Upd2 vesicles (red) in tracheal progenitors (DAPI) in control (**K**), *grasp65RNAi* (**L**), and *lbmRNAi* (**M**) flies. Dashed lines outline tracheal progenitors. (**N**) Bar graph plots the number of Upd2-mCherry-containing vesicles in progenitors. Error bars represent SEM, *n* = 4. Scale bar: 20 μm (**A, B, G–I**), 50 μm (**D, E, K–M**). Genotypes: (**A–G, K**) *UAS-upd2-mCherry/+;lsp2-Gal4/+*; (**H, L**) *UAS-upd2-mCherry/UAS-grasp65RNAi;lsp2-Gal4/+*; (**I, M**) *UAS-upd2-mCherry/+;lsp2-Gal4/UAS-lbmRNAi.*

The online version of this article includes the following figure supplement(s) for figure 6:

**Figure supplement 1.** Microscopic image of a preparation of fly fat body.

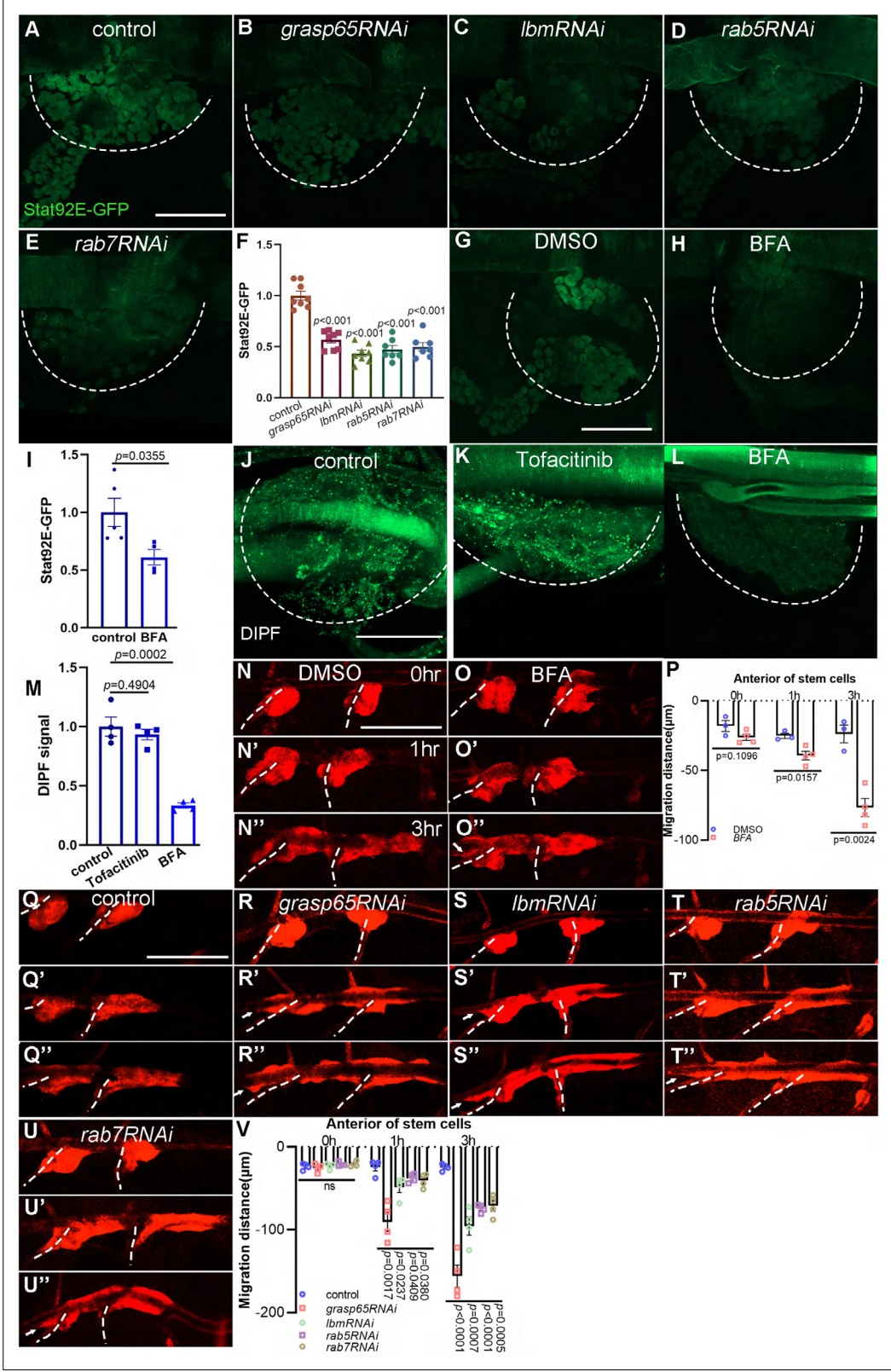

**Figure 7.** The dependence of tracheal progenitors on vesicle trafficking. The expression of Stat92E-GFP in tracheal progenitors of control (**A**), *grasp65RNAi* (**B**), *lbmRNAi* (**C**), *rab5RNAi* (**D**), and *rab7RNAi* (**E**) flies. The progenitors are outlined by dashed lines. (**F**) Bar graph plots the relative expression of Stat92E-GFP. Error bars represent SEM, *n* = 7. The expression of Stat92E-GFP in tracheal progenitors in DMSO-fed (**G**) and BFA-treated (**H**) flies.

*Figure 7 continued on next page*

*Figure 7 continued*

(**I**) Bar graph plots the relative expression of Stat92E-GFP. Error bars represent SEM, *n* = 5. (**J**) The signal of DIPF reporter in tracheal progenitors. (**K**) The effects of Tofacinib (JAK inhibitor) on DIPF reporter in progenitors. (**L**) The effects of Brefeldin A on DIPF reporter in progenitors. Dashed lines outline tracheal progenitors. (**M**) Bar graphs showing the signal of DIPF reporter. Error bars represent SEM, *n* = 4. Migration of tracheal progenitors in DMSO-fed flies (**N–N″**) and BFA-treated flies (**O–O″**). (**P**) Bar graph plots migration distance of anterior movement. Error bars represent SEM, *n* ≥ 3. (**Q–U″**) Migration of tracheal progenitors at 0 hr APF (**Q**), 1 hr APF (**Q′**), and 3 hr APF (**Q″**). The confocal images showing the tracheal progenitors in control (**Q–Q″**), *grasp65RNAi* (**R–R″**), *lbmRNAi* (**S–S″**), *rab5RNAi* (**T–T″**), and *rab7RNAi* (**U–U″**) flies. (**V**) Bar graph plots the migration distance of anterior movement. Error bars represent SEM, *n* = 4. Scale bar: 50 μm (**A–E, G, H, J–L**), 200 μm (**N–O″, Q–U″**). Genotypes: (**A, G, H**) *lsp2-Gal4,Stat92E-GFP/+*; (**B**) *UAS-grasp65RNAi/+;lsp2-Gal4,Stat92E-GFP/+*; (**C**) *lsp2-Gal4,Stat92E-GFP/ UAS-lbmRNAi*; (**D**) *UAS-rab5RNAi/+;lsp2-Gal4,Stat92E-GFP/+*; (**E**) *UAS-rab7RNAi/+;lsp2-Gal4,Stat92E-GFP/+*; (**J–L**) *btl-Gal4/UAS-DIPF*; (**N–Q″**) *lsp2-Gal4,P[B123]-RFP-moe/+*; (**R–R″**) *lsp2-Gal4,P[B123]-RFP-moe/UAS-grasp65RNAi*; (**S–S″**) *lsp2-Gal4,P[B123]-RFP-moe/+;UAS-lbmRNAi/+*; (**T–T″**) *UAS-rab5RNAi/+;lsp2-Gal4,P[B123]-RFP-moe/+*; (**U–U″**) *UAS-rab7RNAi/+;lsp2-Gal4,P[B123]-RFP-moe/+*.

The online version of this article includes the following figure supplement(s) for figure 7:

**Figure supplement 1.** The roles of Rabs in transport of Upd2.

Rab5 and Rab7 (*Figure 8K, L*). The presence of Upd2 in Lbm-containing vesicles was also evidenced in S2 cells (*Figure 8—figure supplement 1A–C″*) and co-IP experiment showed that Lbm physically associated with Upd2 in both fat body and S2 cells (*Figure 8M*, *Figure 8—figure supplement 1F*). To further understand the biogenesis of Lbm-containing vesicles that transport Upd2, we conducted electron microscopic analysis of the Lbm-containing vesicles through the expression of an HRP-fused Lbm in the fat body (*Figure 8—figure supplement 1D*). The interaction between Lbm and Upd2 as simulated by Alphafold2 supported their direct association (*Figure 8—figure supplement 1E*). Then, we generated an Lbm chimera tagged with a pH-sensitive GFP variant, pHluorin (*Yoshihara et al., 2005*). PHluorin fluorescence is squelched at the low pH domain such as in intravesicular compartments, but becomes detectable when exposed to the extracellular environment, thus enabling detection of exocytosis and endocytosis. Fat body expressing Lbm-pHluorin produced GFP puncta at the plasma membrane (*Figure 8N*), and the GFP signal was also detected in the trachea, suggesting the reception and internalization of Lbm-containing vesicles by tracheal cells (*Figure 8O*). However, the GFP fluorescence in both fat body and trachea was dramatically decreased by BFA treatment, suggesting that Lbm-containing vesicles are diminished (*Figure 8—figure supplement 1G–J*). Accordingly, the signals of Lbm-pHluorin in both fat body and responding tracheoblasts were apparently compromised when Rab5 or Rab7 was perturbed (*Figure 8P–S*), suggesting that the biogenesis and production of Lbm-containing vesicles depend on Rab-mediated vesicle trafficking. Taken together, these results suggest that fat body-derived Upd2 interacts with Rab-mediated endocytic trafficking system to control the disciplined movement of tracheal progenitors.

## Discussion

Resident stem cells and progenitors are mobilized to regenerate damaged or degenerated tissue. Despite the large distance between the niche where stem cells interact with their microenvironments and the destination for reconstruction, their commitment to a stereotyped track implicates sophisticated mechanism that controls disciplined migration as stem cells are activated and move out of the niche. While primary inducers expressed by damaged tissues are in play to coordinate the newly generated architecture with the degenerated counterpart (*Chen and Krasnow,*

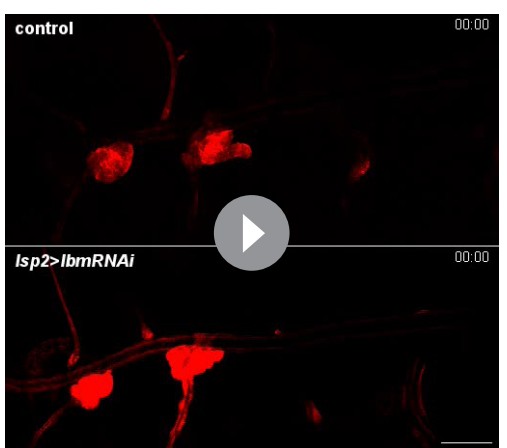

**Video 6.** The movement of tracheal progenitors in control and *lbmRNAi* flies. Scale bar: 100 μm. Genotypes: *lsp2-Gal4,P[B123]-RFP-moe/+* (control) and *lsp2-Gal4,P[B123]-RFP-moe/+;UAS-lbmRNAi/+*. https://elifesciences.org/articles/100037/figures#video6

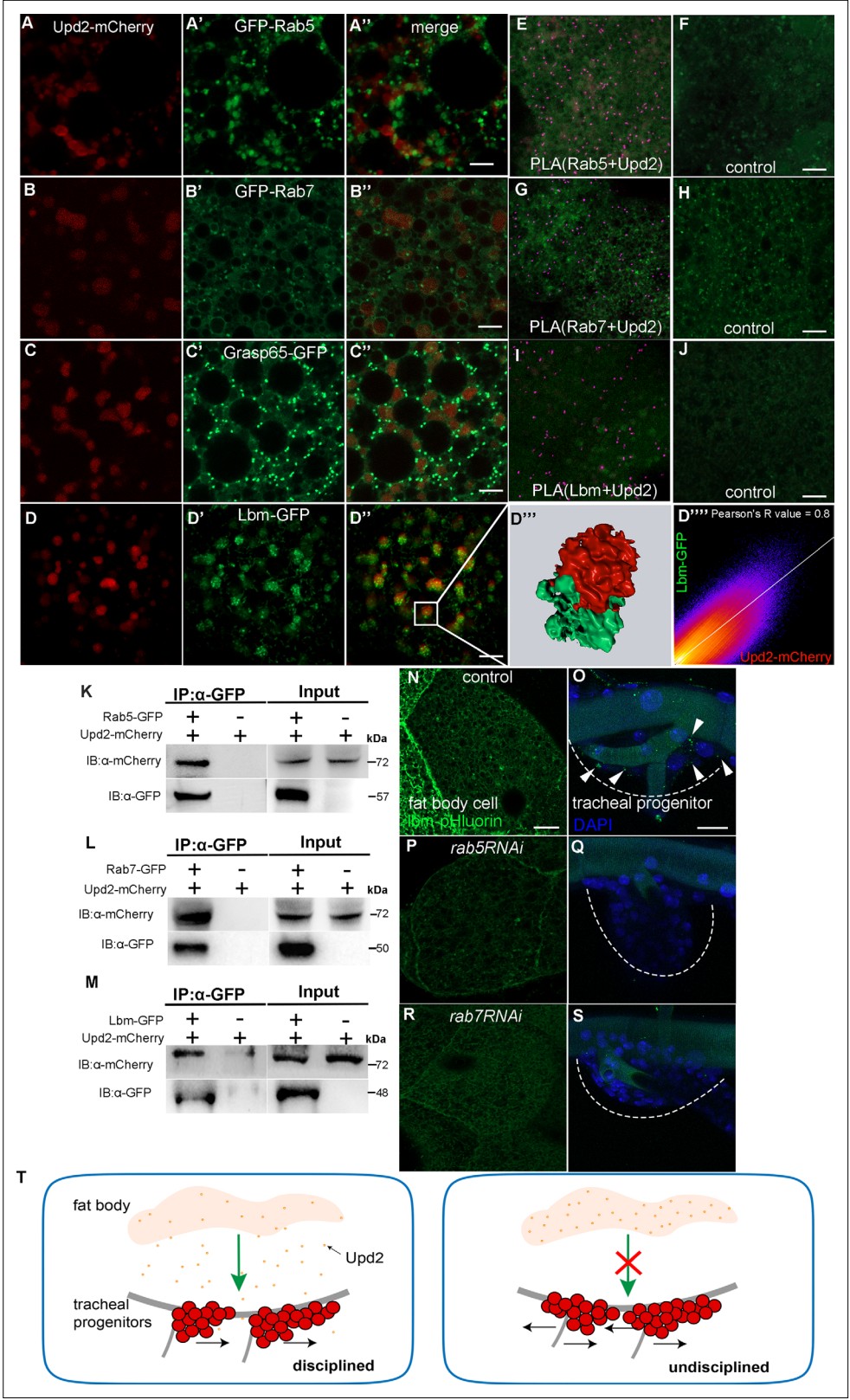

**Figure 8.** The roles of endocytic trafficking system in the transport of Upd2. (**A–D′′′**) The confocal images showing the colocalization between Upd2 (red) (**A–D**) and Rab5 (GFP) (**A′**), Rab7 (GFP) (**B′**), Grasp65 (GFP) (**C′**), or Lbm (GFP). (**A′′, B′′, C′′, D′′**) Merged images. (**D′′′**) 3D high-magnification view of the boxed inset in D′′. (**D′′′′**) The Pearson's correlation coefficient depicting colocalization between Lbm and Upd2 in fat body cells. The PLA (proximity

*Figure 8 continued on next page*

*Figure 8 continued*

ligation assay) assay showing the interaction between Upd2 and Rab5 (**E, F**), Rab7 (**G, H**), or Lbm (**I, J**). Co-immunoprecipitation assay showing physical interaction between Upd2 and Rab5 (**K**), Rab7 (**L**), or Lbm (**M**) in larval fat body. (**N–T**) The expression of Lbm-pHluorin in larval fat body and progenitors of control (**N, O**), *rab5RNAi* (**P, Q**), and *rab7RNAi* (**R, S**) flies. Dashed lines outline tracheal progenitors. Arrowheads point to Lbm-pHluorin puncta. DAPI signal indicates nuclei. (**T**) Schematic diagram depicting Upd2-operated disciplined migration of tracheal progenitors. Scale bar: 5 µm (**A–D", E–J**), 10 µm (**N, P, R**), 20 µm (**O, Q, S**). Genotypes: (**A–A", E, F**) *UAS-upd2-mCherry/+;lsp2-Gal4/UAS-GFP-rab5*; (**B–B", G, H**) *UAS-upd2-mCherry/+;lsp2-Gal4/UAS-GFP-rab7*; (**C–C"**) *UAS-upd2-mCherry/UAS-grasp65-GFP;lsp2-Gal4/+*; (**D–D"", I–L**) *UAS-upd2-mCherry/UAS-lbm-GFP;lsp2-Gal4/+*; (**N, O**) *UAS-lbm-pHluorin/+;lsp2-Gal4/+*; (**P, Q**) *UAS-lbm-pHluorin/UAS-rab5RNAi;lsp2-Gal4/+*; (**R, S**) *UAS-lbm-pHluorin/UAS-rab7RNAi;lsp2-Gal4/+*.

The online version of this article includes the following source data and figure supplement(s) for figure 8:

**Source data 1.** Original files for western blot analysis displayed in *Figure 8K–M*.

**Source data 2.** PDF file containing original western blots for *Figure 8K–M*, indicating the relevant bands and treatments.

**Figure supplement 1.** Vesicular transport of Upd2.

**Figure supplement 1—source data 1.** Original files for western blot analysis displayed in *Figure 8—figure supplement 1F*.

**Figure supplement 1—source data 2.** PDF file containing original western blots for *Figure 8—figure supplement 1F*, indicating the relevant bands and treatments.

---

*2014*), the present study elucidates an integral role of Upd2–JAK–STAT pathway in regulating the expression of polarity-related genes and maintaining the disciplined migration of tracheal progenitors (*Figure 8T*). The transport of Upd2 from fat body to trachea suggests intensive inter-organ communication during the migration of tracheal progenitors.

Several possibilities could account for the JAK/STAT-dependent polarity. The signaling components of JAK/STAT pathway could exhibit polarized localization (*Sotillos et al., 2008*). Alternatively, this signaling may activate genes controlling cell polarity and adhesion (*Mallart et al., 2024*; *Tsurumi et al., 2011*). Functional interplay between JAK/STAT signaling and cell polarity has been observed in various contexts (*Chatterjee et al., 2023*; *Zeidler et al., 1999*). Our results support a role of JAK/STAT signaling in promoting expression of genes with established roles in planar polarity, which may hallmark the route for the migration of the tracheal progenitor cells. Epithelial cells exhibit two aspects of polarity: apical–basal polarity and PCP. The latter refers to the collective alignment of cell polarity within the plane of an epithelial sheet (*Zallen, 2007*). Molecularly, PCP is generated by the asymmetry of a group of proteins (PCP proteins) that mediate communication between neighboring cells (*Barlan et al., 2017*; *Matis and Axelrod, 2013*; *Williams et al., 2022*). In *Drosophila*, the components of PCP are considered to group functionally into two core modules. The seven-pass transmembrane protein Frizzled (Fz), the cytosolic proteins Dishevelled (Dsh), Diego (Dgo), the four-pass transmembrane protein Strabismus (Stbm, also known as Van Gogh (Vang)) and the cytosolic protein Prickle (Pk) belong to the first module. The second module consists of Fat (Ft; also known as cadherin-related tumor suppressor), Dachsous (Ds), Four-jointed (Fj) and Atro (a transcription repressor) (*Peng and Axelrod, 2012*). Intensive functional interplay occurs between these two modules (*Ayukawa et al., 2014*). Aberrant activity of the core PCP proteins leads to misoriented hairs and complex swirling patterns (*Ma et al., 2003*; *Wong and Adler, 1993*). In addition to arrangement of epithelial appendages, PCP pathway is also required for collective and directed cell movements (*Muñoz-Soriano et al., 2012*). The migratory cell cohort is polarized into 'pioneer' cells that lead the trailing followers (*Vitorino and Meyer, 2008*).

Our data indicate that expression of Ds, Fj, Fz, Stan, and Fat2, core components or regulators of PCP, depends on JAK/STAT pathway. The interaction between atypical cadherin Fat (Ft) and its ligand, Dachsous (Ds) directs core protein asymmetry (*Strutt and Strutt, 2021*; *Yang et al., 2002*). Phenotypically, aberrancy of PCP protein abundance, either excessive core protein or deficit in expression gradients, gives rise to similar morphological abnormality (*Adler et al., 2000*; *Casal et al., 2002*; *Taylor et al., 1998*; *Tree et al., 2002*). Consistent with this notion, gain-of-function of Fj or Ds phenocopies that of perturbation of PCP proteins. JAK/STAT signaling promotes the expression of Ds, but

reduces Ft expression (*Figure 4*). Thereby, perturbation of JAK/STAT signaling disrupts the Ds–Ft system.

A precedent for JAK/STAT signaling in directional cell movement is border cell migration from anterior to posterior compartment during *Drosophila* oogenesis. Migration of the border cells is guided by a gradient of PDGF and VEGF chemokines (*Duchek et al., 2001*). Loss of either *hop* (encoding JAK) or *stat* in the border cells impinges their recruitment into the cluster and subsequent migration (*Silver et al., 2005*). Our results suggest that JAK/STAT signaling does not serve as a guidance cue for tracheal progenitors, but rather directs the directionality of cell movement. The downstream PCP components may contribute to either polarity of progenitors or cell–cell interactions between the progenitors and tracheal cells that they track along. It remains unknown how individual progenitor cells perceive directional information and convert it into group choreography.

We identified the fly fat body as the major source for the JAK/STAT signaling ligand, Upd2 production. Fat body is functionally equivalent to the mammalian liver which stores proteins, lipids and sugars and functions as an energy reservoir (*Li et al., 2019*). It supplies proteins and/or hormones that are utilized by other organs, and thereby serves as an interchange center to disperse systemic hormonal and nutritional signals. For instance, it generates collagen IV to decorate imaginal discs and produces xanthine dehydrogenase for eye pigmentation (*Pastor-Pareja and Xu, 2011*; *Reaume et al., 1989*). The transport between fat body and trachea has been reported on a secreted chitin deacetylase, Serpentine (Serp), which is expressed by fat body and contributes to tracheal morphogenesis (*Dong et al., 2014*). Our results reveal that fat body also signals to regulate the disciplined migration of tracheal progenitors through the dispersion of Upd2 cytokines. These studies collectively suggest that fat body orchestrates systemic tissue growth and patterning and that metabolic regulation is critical for adult stem cells.

Proteins that are locally produced can execute systemic function in distant organs. A possible route of transport is through the hemolymph or bloodstream and taken up by the target tissues. The signaling proteins such as cytokines can be packaged in extracellular vesicles with various dimensions (*Buzas, 2023*; *Javeed et al., 2021*). A precedent of vesicular transport of signaling molecules is reported in migrasomes whose diameter exceeds 500 nm (*Jiang et al., 2019*).

These extracellular vesicles mediate cell-to-cell communication (*Colombo et al., 2014*), perhaps at a distance (*Hood et al., 2011*) and even traverse between organs (*Corrigan et al., 2014*). The vesicular Upd2 is able to signal at recipient cells, suggesting that the activity of Upd2 is preserved in the vesicle, and it is released upon vesicle fusion. Compared with conventional extracellular vesicles such as exosomes, the Upd2-containing vesicles possess larger dimension. Its production and trafficking depend on GRASP-mediated unconventional secretion and interaction with Lbm. Lbm belongs to the tetraspanin protein family that contains four transmembrane domains. The mammalian homologs of tetraspanins, CD9, CD63, CD81, or CD37 are principal constituents of extracellular vesicles. The Lbm-containing vesicles are regulated by GRASP-mediated secretion and are sensitive to pharmacological inhibition of EV transport.

It has been proposed that tetraspanins facilitate regeneration and wound healing in cultural cells and single-cell plasma membrane. Tetraspanin-enriched macrodomains are assembled into a ring-like structure (*Huang et al., 2022*), which is recruited to large membrane wounds and promotes membrane repair (*Wang et al., 2022*). The present study adds another dimension to the roles of tetraspanin proteins in tissue regeneration which can be ascribed to transport of signaling proteins and modulation of stem cells.

# Materials and methods

**Key resources table**

| Reagent type (species) or resource | Designation | Source or reference | Identifiers | Additional information |
|---|---|---|---|---|
| Gene (*Drosophila melanogaster*) | upd2 | THFC | THU1331, THU1288 | |
| Gene (*Drosophila melanogaster*) | dome | THFC | THU0574, THU5825 | |

*Continued on next page*

*Continued*

| Reagent type (species) or resource | Designation | Source or reference | Identifiers | Additional information |
|---|---|---|---|---|
| Gene (*Drosophila melanogaster*) | hop | THFC | THU5762, TH201501042.S | |
| Gene (*Drosophila melanogaster*) | stat92E | THFC | THU0573, THU1915 | |
| Gene (*Drosophila melanogaster*) | lbm | THFC, BDSC | THU2602, BDSC27278 | |
| Gene (*Drosophila melanogaster*) | grasp65 | THFC | TH04282.N, THU1429 | |
| Gene (*Drosophila melanogaster*) | rab5 | THFC | TH02192.N, THU0671 | |
| Gene (*Drosophila melanogaster*) | rab7 | THFC | TH02539.N, THU2437 | |
| Gene (*Drosophila melanogaster*) | fj | THFC | THU201500988.S, THU1538 | |
| Gene (*Drosophila melanogaster*) | fat2 | THFC, VDRC | THU4120, VDRC27113 | |
| Gene (*Drosophila melanogaster*) | yurt | THFC, VDRC | THU1740, VDRC28674 | |
| Gene (*Drosophila melanogaster*) | crb | THFC | THU2783, THU5212 | |
| Gene (*Drosophila melanogaster*) | scb | THFC | THU3905, THU2707 | |
| Gene (*Drosophila melanogaster*) | ds | THFC, VDRC | THU2846, VDRC36219 | |
| Gene (*Drosophila melanogaster*) | ft | THFC, VDRC | TH201500989.S, VDRC9396 | |
| Genetic reagent (*Drosophila melanogaster*) | 10×Stat92E-GFP | THFC | THJ0273 | |
| Genetic reagent (*Drosophila melanogaster*) | Stat92E-GFP | BDSC | BDSC:38670 | |
| Genetic reagent (*Drosophila melanogaster*) | UAS-grasp65-GFP | BDSC | BDSC:8507 | |
| Genetic reagent (*Drosophila melanogaster*) | Ds::GFP | BDSC | BDSC:59425 | |
| Genetic reagent (*Drosophila melanogaster*) | UAS-GFP-Rab7 | BDSC | BDSC:9779 | |
| Genetic reagent (*Drosophila melanogaster*) | UAS-GFP-Rab5 | BDSC | BDSC:24616 | |
| Genetic reagent (*Drosophila melanogaster*) | hop$^{Tum}$/FM7C | BDSC | BDSC:8492 | |
| Genetic reagent (*Drosophila melanogaster*) | stat92E$^F$ | BDSC | BDSC:24757 | |

*Continued on next page*

*Continued*

| Reagent type (species) or resource | Designation | Source or reference | Identifiers | Additional information |
|---|---|---|---|---|
| Genetic reagent (*Drosophila melanogaster*) | dome^G0264 | Kyoto | Kyoto:111866 | |
| Genetic reagent (*Drosophila melanogaster*) | Yurt::GFP | VDRC | VDRC:318067 | |
| Genetic reagent (*Drosophila melanogaster*) | Crb::GFP | VDRC | VDRC:318384 | |
| Genetic reagent (*Drosophila melanogaster*) | Ft::GFP | VDRC | VDRC:318477 | |
| Genetic reagent (*Drosophila melanogaster*) | Fj::GFP | VDRC | VDRC:318457 | |
| Genetic reagent (*Drosophila melanogaster*) | UAS-ft | This paper; **Brittle et al., 2010** | | |
| Genetic reagent (*Drosophila melanogaster*) | UAS-ds | This paper; **Brittle et al., 2012** | | |
| Genetic reagent (*Drosophila melanogaster*) | Fat2::GFP | This paper; **Barlan et al., 2017** | | |
| Cell line (*D. melanogaster*) | S2 | CCTCC | GDC#0138 | Verified by DNA barcoding; without mycoplasma contamination |
| Cell line (*Homo sapiens*) | SKOV3 | ATCC | HTB-77 | Verified by STR genotyping; without mycoplasma contamination |
| Antibody | anti-GFP (Mouse monoclonal) | Abclonal | Cat# AE012 | WB (1:1000) |
| Antibody | anti-mCherry (Mouse polyclonal) | Abclonal | Cat# AE002 | WB (1:1000) |
| Antibody | Alexa Fluor 488 | Abclonal | Cat# AS053 | IF (1:200) |
| Antibody | Alexa Fluor 555 | Abclonal | Cat# AS007 | IF (1:200) |
| Antibody | Phalloidin Alexa Fluor 640 | Biotum | Cat# 00050 | IF (1:50) |
| Antibody | anti-tubulin (Rabbit polyclonal) | Baoke | Cat# BK7010 | WB (1:5000) |
| Antibody | anti-GFP (Rabbit polyclonal) | Invitrogen | Cat# A11122 | IF (1:400) |
| Antibody | HRP-conjugated Streptavidin | Proteintech | Cat# SA00001 | WB (1:5000) |
| Sequence-based reagent | α-tubulin84b _F | This paper | PCR primers | CACACCACCCTGGAGCATTC |
| Sequence-based reagent | α-tubulin84b _R | This paper | PCR primers | CCAATCAGACGGTTCAGGTTG |
| Sequence-based reagent | upd2_F | This paper | PCR primers | TCAATCCGTATCGCGGTCTG |
| Sequence-based reagent | upd2_R | This paper | PCR primers | AGAAGAGTCGCAGGTTGTGG |
| Sequence-based reagent | ds_F | This paper | PCR primers | ACAACCGAACTCGAACCGAA |
| Sequence-based reagent | ds_R | This paper | PCR primers | AGTAGCATCACACACAAGTGA |

*Continued on next page*

*Continued*

| Reagent type (species) or resource | Designation | Source or reference | Identifiers | Additional information |
|---|---|---|---|---|
| Sequence-based reagent ft_F | | This paper | PCR primers | CTGGATCGAGAGCAGCAGAG |
| Sequence-based reagent ft_R | | This paper | PCR primers | GACGGTAAATTCTCGCGCAC |
| Sequence-based reagent fj_F | | This paper | PCR primers | ATTACTCAAGCGGTTGGGGG |
| Sequence-based reagent fj_R | | This paper | PCR primers | CGGTTCCTGTTCCTGTCTCC |
| Sequence-based reagent fat2_F | | This paper | PCR primers | TATCTGCGCCCATACGCATT |
| Sequence-based reagent fat2_R | | This paper | PCR primers | TCTCATCGGCCTTGCTTTGT |
| Sequence-based reagent yurt_F | | This paper | PCR primers | GGTCAGCTCAGGGTGACTATC |
| Sequence-based reagent yurt_R | | This paper | PCR primers | ATTGGTAAGCTTGGCGTTGC |
| Sequence-based reagent crb_F | | This paper | PCR primers | CAGCAGTGTTTGAACGGTGG |
| Sequence-based reagent crb_R | | This paper | PCR primers | AGGCAGTGACCAATGGGG |
| Peptide, recombinant protein | Anti-FLAG M2 Magnetic Beads | Millipore | Cat# 8823 | |
| Commercial assay or kit | RNeasy Micro Kit | QIAGEN | Cat# 74004 | |
| Commercial assay or kit | SMART-Seq v4 Ultra low input RNA Kit | Takara | Cat# 634889 | |
| Commercial assay or kit | AMPure XP | Beckman Coulter | Cat# A63882 | |
| Commercial assay or kit | TruePrep DNA Library Prep Kit V2 | Vazyme | Cat# TD501 | |
| Chemical compound, drug | BXXP | APEXBIO | Cat# A8012 | |
| Software, algorithm | Fiji/ImageJ | NIH | RRID:SCR_002285 | |
| Software, algorithm | GraphPad Prism 8.0 | GraphPad Software | https://www.graphpad.com/scientific-software/prism/ | |
| Software, algorithm | Zen 3.1 | Zeiss | https://www.zeiss.com.cn/corporate/home.html | |
| Software, algorithm | PCA-flow | Bradski, G.79 | https://www.drdobbs.com/open-source/the-opencv-library | |
| Other | DAPI | VECTASHIELD | Cat# H1200 | |

## Fly lines and husbandry

All flies were reared on normal cornmeal and agar medium at 25°C unless noted. UAS-upd2RNAi (THU1331, THU1288), UAS-domeRNAi (THU0574), UAS-hopRNAi (THU5762), UAS-stat92ERNAi (THU0573), UAS-lbmRNAi (THU2602), UAS-grasp65RNAi (TH04282.N), UAS-rab5RNAi (TH02192.N), UAS-rab7RNAi (TH02539.N), UAS-fjRNAi (THU201500988.S), UAS-fat2RNAi (THU4120), UAS-yurtRNAi (THU1740), UAS-crbRNAi (THU2783), and UAS-scbRNAi (THU3905) were ordered from Tsinghua Stock Center. Stat92E-GFP (BSDC#38670), UAS-grasp65-GFP (BSDC#8507), Ds::GFP (BSDC#59425) were from Bloomington *Drosophila* Stock Center. Yurt::GFP (v318067), Crb::GFP (v318384), Ft::GFP (v318477), Fj::GFP (v318457), UAS-dsRNAi (v36219, THU2846), and UAS-ftRNAi (v9396) were obtained from VDRC. UAS-ft was kindly provided by Dr. Xianjue Ma, UAS-ds was kindly provided by Dr. Xing Wang and UAS-fat2-GFP was from Dr. Shunfan Wu. UAS-GFP-rab5 and UAS-GFP-rab7 were kindly provided by Dr. Xiaohang Yang.

## Plasmid construction and transgenic flies

To generate UAS-upd2-mCherry and UAS-lbm-GFP, UAS-lbm-HRP transgenic flies, the coding sequence of upd2 or lbm was PCR amplified from a fly cDNA library and cloned into a pUAST vector with C-terminal mCherry, GFP, or HRP.

The DIPF reporter was generated by first fusing cpYFP and *Drosophila* FKBP12 and subsequently ligating to the dome cDNA via a GTG linker. The above product was then cloned into a pUAST vector and verified by DNA sequencing, and injected into y[1] M{vasint.Dm}ZH-2A w[*]; P{CaryP}attP2 recipient flies or *w1118* following standard *Drosophila* transformation injection procedures (Core Facility of *Drosophila* Resource and Technology, SIBCB, CAS).

## Cell culture and transfection

S2 cells (CCTCC, GDC#0138) were grown in Schneider *Drosophila* Medium (Gibco, #21720024) supplemented with 10% (vol/vol) fetal bovine serum (FBS, Gibco, #10099141C) and 1% (vol/vol) penicillin–streptomycin (Pen/Strep, Life Technologies) at 28°C with 0.2% $CO_2$. S2 cells were confirmed by DNA barcoding and verified to be mycoplasma-free using the Mycoplasma Stain Assay Kit. Transfection was conducted with 5 µg plasmids (act-GAL4, UAS-lbm-GFP, UAS-upd2-mCherry) using Effectene Transfection Reagent (QIAGEN, #301425).

SKOV3 cells (ATCC, HTB-77) were cultured in DMEM medium (CR#12800) containing 10% FBS (Gibco, #10099141C) in an incubator with 5% CO2 at 37°C. The SKOV3 cells were confirmed by STR genotyping and verified to be mycoplasma-free using the Mycoplasma Stain Assay Kit. Cells with ~80% confluency were infected with lentivirus loaded with siRNAs for gene knockdown or gRNAs for knockout, and the medium was replaced 24 hr post-infection. 72 hr after lentivirus infection, brightfield images were taken every 5 min for 12 hr using a confocal microscope.

## Quantitative reverse transcription PCR

Larval or pupal trachea were dissected in cold PBS, and then transferred to RNA extraction reagent (AG21101). Next, reverse transcription was performed using qPCR RT Mix with gDNA Remover reagent (AG11706). qPCR was performed using the Universal SYBR Select Master Mix (AG11701) with a Bio-Rad system. The foldchange of target gene expression was normalized to that of α-tubulin. The primers are listed in Key Resources Table.

## Western blotting and co-immunoprecipitation

Total protein was extracted from cells or tissues by RIPA buffer supplemented with a protease inhibitor cocktail (Merck, #11836170001) and phenylmethanesulfonyl fluoride (Beyotime, #ST507), separated by 10% SDS–PAGE gels and transferred to PVDF membrane (Millipore, #IPVH00010). Blots were detected with an ECL Western Blotting detection system (Bio-Rad). For co-immunoprecipitation, lysates of larval fat body or transfected S2 cells were incubated overnight at 4°C with protein A Magnetic beads (Thermo Scientific, #2736141) pre-coated with GFP antibody (Invitrogen, #A11122). Immunoprecipitates were eluted in SDS-containing loading buffer for subsequent immunoblotting analysis. Antibodies for immunoblotting include: α-tubulin (Baoke, #BK7010), α-HRP-conjugated streptavidin (Proteintech, #SA00001-0), α-GFP (Abclonal, #AE012, 1:1000), and α-mCherry (Abclonal, #AE002, 1:1000).

## Proximity ligation assay

PLA was carried out with Duolink In Situ Detection Reagents Far Red (Sigma-Aldrich, #DUO92013) according to manufacturer's instructions, using the probes anti-rabbit PLUS (Sigma-Aldrich, #DUO92002) and anti-mouse MINUS (Sigma-Aldrich, #DUO92004). Briefly, larval fat body was dissected and fixed in 4% formaldehyde. The animals not expressing upd2-mCherry served as controls. After permeabilization, the samples were incubated with primary antibodies overnight at 4°C. Then, the samples were washed with PLA buffer A, hybridized with PLA probes, ligated, and amplified. Samples were washed twice with PLA buffer B (Sigma-Aldrich, #DUO82049) and fluorescence images were taken with an LSM Zeiss 900 inverted confocal laser scanner microscope.

## Immunofluorescence

Trachea from white pupae (0 hr APF) were dissected in PBS and fixed with 4% formaldehyde for 25 min at room temperature. After washes, trachea samples were permeabilized with 1% Triton X-100 in PBS, and then blocked in 10% goat serum. Incubation with primary antibody (GFP, 1:400; lacZ, 1:40) was performed at 4°C with gentle rotation for overnight. Then, the samples were incubated with secondary antibodies conjugated to Alexa Fluor 488 or 555 (1:200) and Phalloidin (1:50) for 2 hr.

After washing, samples were mounted in antifade mounting medium with DAPI (VECTASHIELD) and imaged under an LSM Zeiss 900 inverted confocal laser scanner microscope.

## Live imaging of pupal trachea stem cells

White pupae (0 hr APF) were briefly washed in double distilled water and mounted in halocarbon oil 700 (Sigma, #H8898). The pupae were positioned with forceps to bring a single DT of the trachea up for optimal imaging of Tr4 and Tr5 metameres. Then, pupae were immobilized by a 22 × 30 mm No. 1.5 high precision coverslip spaced by vacuum grease. Time-lapse images were captured by an LSM Zeiss 900 inverted confocal laser scanner microscope. For migration distance measurement, we took sequential snapshots of the moving progenitors of pupae staged at 0, 1, 2, and 3 hr APF. The migration distance was measured as the distance from the starting position (the junction of TC and DT) to the leading edge of progenitor groups. The migration velocity was calculated by $v = d$ (micrometer)$/t$ (min).

## RNA sequencing of tracheal progenitors

Total RNA was isolated from the Tr4 and Tr5 metamere progenitors dissected from 1 hr APF pupae using RNeasy Micro Kit (QIAGEN, #74004). SMART-Seq v4 Ultra low input RNA Kit (Takara, #634889) was used for first- and second-strand cDNA synthesis and double-stranded cDNA end repair. Double-stranded cDNAs were cleaned using AMPure XP (Beckman Coulter, #A63882). Then cDNAs were subjected to tagmentation and ligation to adaptors to generate the sequencing libraries using True-Prep DNA Library Prep Kit V2 for Illumina kit (Vazyme, #TD501). The quality and concentration of the libraries were assessed using the Agilent High Sensitivity DNA Kit and Bioanalyzer 2100 (Agilent Technologies) and submitted to 150 bp paired-end high throughput sequencing using Hiseq4000 (Illumina).

Analysis of RNA-seq data was performed using a computer system equipped with multiple processors. Clean reads were mapped to the *Drosophila* genome sequence using Hisat2 with default parameters. Successfully mapped reads were counted using FeatureCounts. Differential gene expression analysis was performed using the DESeq2 package. Adjusted p-value <0.05 was used as the threshold to identify the DEGs. Gene ontology and KEGG pathway enrichment analyses for the DEGs were conducted using the Database for Annotation, Visualization and Integrated Discovery (DAVID).

## Chromatin immunoprecipitation

Third instar larval trachea from Stat92E-Flag (BDSC, #38670) were fixed in 1% formaldehyde. The fixation reaction was terminated by adding glycine (125 mM). Trachea were washed and resuspended in lysis buffer, and sonicated to generate 200–600 bp DNA fragments. Procedures of immunoprecipitation and ChIP sequencing library construction were as previously described (*Li et al., 2022*). Anti-FLAG M2 Magnetic Beads (Millipore, #8823) were used for enriched DNA binding to transcription factor Stat92E.

Immunoprecipitated DNA was subjected to next-generation sequencing using the Epicenter Nextera DNA Sample Preparation Kit or to real-time PCR. Library construction was performed using the High Molecular Weight tagmentation buffer, and tagmented DNA was linearly amplified by PCR. The libraries were then sequenced on a Novaseq according to the manufacturer's standard protocols. The sequences were processed using *Fastqc* and low-quality bases and adaptor contamination were trimmed by *cutadapt*. Filtered reads were mapped to *Drosophila* genome using *BWA mem* algorithm. Peaks were called using macs2 *callpeak* (*Zhao et al., 2019*) and plotted using pyGenomeTracks. GO analysis of biological processes was conducted by DAVID.

## Cell-surface proteomics of fly trachea

Trachea from white pupae were dissected in pre-cooled Schneider Medium (Gibco) and collected in 1.5 ml low-binding tube (Axygen) containing 500 µl Schneider Medium. The samples were washed with 500 µl fresh medium and incubated with 100 µM BXXP (APEXBIO, #A8012) for 1 hr on ice with occasional pipetting. Labeling reaction was initiated by adding 1 mM (0.03%) $H_2O_2$ to the sample-containing medium and proceeded for 7 min at room temperature. The reaction was immediately quenched by five thorough washes with PBS containing 10 mM sodium ascorbate (Aladdin, #S105024) and 5 mM Trolox (APEXBIO, #C3183). For biochemical characterization or proteomic sample preparation, the

quenching solution was drained, and the trachea in minimal residual quenching solution were quickly frozen in liquid nitrogen and stored at 80°C. LC–MS/MS analysis was performed using a Q Exactive HF-X instrument (Thermo Fisher) coupled with Easy-nLC 1200 system. The acquired MS raw data were processed using MaxQuant version 2.0.1.0 (Max Planck Institute of Biochemistry, Germany). Label-free quantification was set with a default parameter and iBAQ was selected.

## Transmission electron microscopy

Fat body of third instar larvae were dissected and fixed in 0.12 M Na-cacodylate buffer (pH 7.4) containing 2.5% glutaraldehyde for 1 hr on ice. Then the samples were rinsed in 0.12 M Na-cacodylate buffer (6 × 5 min, on ice). The dissected fat body were pre-incubated with DAB (10 mg/20 ml) in 0.12 M Na-cacodylate buffer (containing 0.1% saponin) for 30 min with agitation in the dark. Then 30% $H_2O_2$ was quickly mixed in DAB solution to a 0.03% vol/vol concentration and reacted for 30 min at RT. The fat body were transferred to 0.12 M Na-cacodylate buffer (6 × 5 min, RT). To increase the electron density of the HRP/DAB product, samples were transferred into 0.01% $OsO_4$ in 0.12 M Na-cacodylate buffer (pH 7.4) for 10 min at RT, then rinsed in 0.12 M Na-cacodylate buffer (3 × 10 min, on ice). 0.1% thiocarbohydrazide in 0.12 M Na-cacodylate buffer (pH 7.4) was used for 10 min at RT, then rinsed in 0.12 M Na-cacodylate buffer (3 × 10 min, on ice). After post-fixation for 1 hr at RT with 1% $OsO_4$ in 0.12 M Na-cacodylate buffer (pH 7.4), the samples were rinsed with MilliQ water (3 × 5 min, RT) and dehydrated in a series of 15 min with 10%, 30%, 50%, 70%, 90%, and 100% (3×) ethanol. Infiltration was conducted at RT with a mixture of acetone and resin 1:1 for 1.5 hr, 1:2 for 3 hr, and 1:3 overnight. The tissues were then dissected from the carcasses and placed in block molds filled with resin for hardening at 60°C during 48 hr. 70 nm ultrathin section from the hardened blocks were cut on a Leica EM UC7 ultramicrotome using an Ultra 45° diamond knife and imaged in a thermos scientific Talos L120C electron microscope.

## Numeration of tracheal progenitors

The pupae at indicated stages were dissected and then fixed with 4% PFA for 25 min. The nuclei were labeled by DAPI. The samples were imaged by an LSM Zeiss 900 inverted confocal laser scanner microscope. The number of progenitor cells was scored from image stacks.

## Optic flow analysis
### Motion correlation

Time-lapse images of trachea progenitor cells were captured every 5 min over a total duration of 2 hr using an LSM Zeiss 900 microscope. The movies and images were subjected to a three-step motion collection using ImageJ (*Schneider et al., 2012*) as follows:

1. Images were transformed to gray scale.
2. Automated stabilization was performed on gray images by image stabilizer.
3. The feature points were mapped and affine transformation was applied using big warp (*Bogovic et al., 2015*).

### Optical flow

Optical flow represents the pattern of motion of pixels in a sequence of images. Between two consecutive frames $I(x, y, t)$ and $I(x, y, t + \Delta t)$, the optical flow vector $v = (v_x, v_y)$ represents the motion of pixels during this time. The optical flow constraint equation is shown below:

$$I_x v_x + I_y v_y + I_t = 0$$

We adopted the PCA-flow algorithm (*Bradski, 2000*) in openCV library (cv::optflow::OpticalFlowP-CAFlow) in which the sparse optical flow vectors within each small region of the image sequence are calculated before training optical flow fields via principal component analysis (PCA) (*Wulff and Black, 2015*). The vectors were assembled to generate a smooth vector field of optic flow using the learning linear models of flow. The PCA-flow was validated for the efficiency and robustness.

## Variance in 1D axis

We developed a robust estimator to evaluate the variance of optic flow projection along the 1D migration axis between samples. The vector length is normalized by a frame-specific normalization factor κ to fit with scale of optic flow computed among different frames.

$$\kappa = Q_{0.9} \left\{ \|v\|_2 \,\big|\, v \in vectorfield \right\}$$

where $Q$ is the quantile function. The variance is computed as $\mathrm{Var}(\{\|v\|_2 > Q_{0.2} | v \in vectorfield\})$ and scaled to 0–1 by $\mathrm{Var}(x) / (\mathrm{Var}(x) + \mathrm{Var}(y))$.

## Random variable

The direction of optic flow in each volume was assigned to 'left' or 'right'. Then the distribution of the binarized directions is modeled as a Bernoulli random variable $X\,Bernoulli\,(p)$ with PMF

$$P(X = x) = \begin{cases} p & \text{if } x = 1, \\ 1 - p & \text{if } x = 0. \end{cases}$$

## Binary entropy

The entropy for Bernoulli random variable $X\,Bernoulli\,(p)$ is defined as

$$H_{binary}(X) = -p\,log\,(p) - (1 - p)\,log\,(1 - p)$$

The entropy evaluates the information contained in the random variable (also called uncertainty). In this case, when $p = 0.5$, it reaches the maxima 1; if the variable is determinate (i.e. $p = 0$ or 1), the entropy is zero. More directed cell migration leads to a lower entropy in optic flow since the certainty is high for the migration direction. We estimated $p$ by computing ratio of left and right direction

$$p = \frac{\#left}{\#left + \#right}$$

## Image and statistical analysis

Confocal z-stack images were analyzed to extract information of fluorescent intensity of stat92E-GFP and DIPF, and the number of Upd2-mCherry-containing vesicles. z slices of fluorescent intensity for Ds, Ft, and Fj were measured. The number of particles for DIPF and Upd2-mCherry-containing vesicles was analyzed using ImageJ. All statistical analysis was conducted using GraphPad Prism 8.0. Mean and SEM were shown. Unpaired *t*-tests with Benjamin's correction were used to evaluate statistical significance between groups.

## Acknowledgements

We thank Drs Thomas Kornberg, David Strutt, Xiaohang Yang, Markus Affolter, Mark Krasnow, Stefan Luschnig, Kai Yuan, Xing Wang, Zhouhua Li, Xianjue Ma, Sarah Bray, Yiming Zheng, and Shunfan Wu for generously providing reagents; Core Facility of *Drosophila* Resource and Technology, SIBCB, CAS for injection service; Bloomington *Drosophila* Stock Center, Kyoto Stock Center, Vienna Stock Center, Tsinghua Stock Center for fly stocks; Developmental Studies Hybridoma Bank for antibodies; all members of Huang lab and Kornberg lab for discussions and constructive suggestions. This work has been financially supported by NSFC92168101, NSFC32070784, and Thousand Young Talent Program to HH, NSFC32000574 to HGW, National Key R&D Program of China (2022YFF0608402) and Chinese National Natural Science Funds (22374128) to BY and NSFC32300699, LQ24C120001 and Postdoctoral Fellowship Foundation 2023M733098 to YL.

# Additional information

## Funding

| Funder | Grant reference number | Author |
|---|---|---|
| National Natural Science Foundation of China | NSFC92168101 | Hai Huang |
| National Natural Science Foundation of China | NSFC32070784 | Hai Huang |
| Thousand Young Talents Program of China | | Hai Huang |
| National Natural Science Foundation of China | NSFC32000574 | Honggang Wu |
| National Key Research and Development Program of China | 2022YFF0608402 | Bing Yang |
| National Natural Science Foundation of China | 22374128 | Bing Yang |
| National Natural Science Foundation of China | NSFC32300699 | Yue Li |
| Postdoctoral Research Foundation of China | 2023M733098 | Yue Li |
| Zhejiang Association for Science and Technology | LQ24C120001 | Yue Li |

The funders had no role in study design, data collection, and interpretation, or the decision to submit the work for publication.

## Author contributions

Pengzhen Dong, Validation, Investigation, Visualization, Methodology, Writing – review and editing; Yue Li, Yuying Wang, Validation, Investigation, Visualization; Qiang Zhao, Data curation, Investigation; Tianfeng Lu, Data curation, Formal analysis, Investigation; Jian Chen, Validation, Investigation; Tianyu Guo, Formal analysis, Methodology; Jun Ma, Honggang Wu, Supervision, Writing – original draft, Writing – review and editing; Bing Yang, Supervision, Methodology, Project administration; Hai Huang, Conceptualization, Supervision, Funding acquisition, Writing – original draft, Writing – review and editing

## Author ORCIDs

Pengzhen Dong ⓘ https://orcid.org/0009-0009-7124-695X
Hai Huang ⓘ https://orcid.org/0000-0003-2331-6238

Reviewer #1 (Public review): https://doi.org/10.7554/eLife.100037.4.sa1
Author response https://doi.org/10.7554/eLife.100037.4.sa2

# Additional files

## Supplementary files

MDAR checklist
Source data 1. Source data for figures.

## Data availability

The mass spectrometry proteomics data have been deposited to the ProteomeXchange Consortium (ProteomeXchangeID number PXD049142) via the iProX partner repository (*Ma et al., 2019*) with project ID IPX0008149000. The RNA-Seq data generated and analyzed in this study have been deposited in the NCBI database under accession number GSE256177. The SMART-Seq data (L3, 0hr APF and 2hr APF) analyzed in this study have been deposited in the NCBI database under accession

number GSE184856. The ChIP-Seq data generated and analyzed in this study have been deposited in the NCBI database under accession number GSE256176.

The following datasets were generated:

| Author(s) | Year | Dataset title | Dataset URL | Database and Identifier |
|---|---|---|---|---|
| Dong P, Li Y, Wang Y, Huang H | 2025 | Drosophila tracheal cell mass spectrum data | https://www.iprox.cn/page/project.html?id=IPX0008149000 | iProX, IPX0008149000 |
| Dong P, Li Y, Wang Y, Huang H | 2024 | Fat body-derived cytokine Upd2 regulates the polarity of *Drosophila* tracheal stem cells [RNA-seq] | https://www.ncbi.nlm.nih.gov/geo/query/acc.cgi?acc=GSE256177 | NCBI Gene Expression Omnibus, GSE256177 |
| Huang H, Zhao Q | 2025 | Drosophila tracheal cell mass spectrum data | https://proteomecentral.proteomexchange.org/cgi/GetDataset?ID=PXD049142 | ProteomeXchange, PXD049142 |

The following previously published datasets were used:

| Author(s) | Year | Dataset title | Dataset URL | Database and Identifier |
|---|---|---|---|---|
| Li Y, Dong P, Guo T, Huang H | 2022 | Genes regulated by Yki play roles in the cell cycle, cell migration and cell adhesion in *Drosophila* | https://www.ncbi.nlm.nih.gov/geo/query/acc.cgi?acc=GSE184856 | NCBI Gene Expression Omnibus, GSE184856 |
| Dong P, Li Y, Wang Y, Huang H | 2025 | Fat body-derived cytokine Upd2 regulates the polarity of Drosophila tracheal stem cells [ChIP-seq] | https://www.ncbi.nlm.nih.gov/geo/query/acc.cgi?acc=GSE256176 | NCBI Gene Expression Omnibus, GSE256176 |

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
