## [Editor Report · eLife Assessment]

This **valuable** study investigates how inter-organ communication between the tracheal stem cells and the fat body plays a key role in the directed migration of tracheal stem cells in *Drosophila* pupae. The evidence supporting the conclusions is **convincing**. The work would be of broad interest to researchers in the fields of developmental biology and cancer biology.

---

## [Referee Report · Reviewer #1 (Public review)]

Summary:

In this manuscript, Dong et al. study the directed cell migration of tracheal stem cells in *Drosophila* pupae. The migration of these cells which are found in two nearby groups of cells normally happens unidirectionally along the dorsal trunk towards the posterior. Here, the authors study how this directionality is regulated. They show that inter-organ communication between the tracheal stem cells and the nearby fat body plays a role. They provide compelling evidence that Upd2 production in the fat body and JAK/STAT activation in the tracheal stem cells plays a role. Moreover, they show that JAK/STAT signalling might induce the expression of apicobasal and planar cell polarity genes in the tracheal stem cells which appear to be needed to ensure unidirectional migration. Finally, the authors suggest that trafficking and vesicular transport of Upd2 from the fat body towards the tracheal cells might be important.

Strengths:

The manuscript is well written. This novel work demonstrates a likely link between Upd2-JAK/STAT signalling in the fat body and tracheal stem cells and the control of unidirectional cell migration of tracheal stem cells. The authors show that hid+rpr or Upd2RNAi expression in a fat body or Dome RNAi, Hop RNAi, or STAT92E RNAi expression in tracheal stem cells results in aberrant migration of some of the tracheal stem cells towards the anterior. Using ChIP-seq as well as analysis of GFP-protein trap lines of planar cell polarity genes in combination with RNAi experiments, the authors show that STAT92E likely regulates the transcription of planar cell polarity genes and some apicobasal cell polarity genes in tracheal stem cells which appear to be needed for unidirectional migration. Moreover, the authors hypothesise and provide some supporting evidence that extracellular vesicle transport of Upd2 might be involved in this Upd2-JAK/STAT signalling in the fat body and tracheal stem cells, which is quite interesting. Overall, the work presented here provides some novel insights into the mechanism that ensures unidirectional migration of tracheal stem cells that prevents bidirectional migration. This might have important implications for other types of directed cell migration in invertebrates or vertebrates including cancer cell migration.

Weaknesses:

It remains somewhat unclear how Upd2 transported in extracellular vesicles would bind to the Dome receptor found on the surface of the tracheal cells? How Upd2 would be released from vesicles to bind Dome extracellularly and activate the JAK-STAT pathway?

---

## [Author Response]

The following is the authors’ response to the previous reviews

**Joint Public Review:**
SummaryIn this manuscript, Dong et al. study the directed cell migration of tracheal stem cells in *Drosophila* pupae. The authors study how the directionality of these cells is regulated along the dorsal trunk. They show that inter-organ communication between the tracheal stem cells and the nearby fat body plays a role in posterior migration. They provide compelling evidence that Upd2 production in the fat body and JAK/STAT activation in the tracheal stem cells play a role. Moreover, they show that JAK/STAT signalling might induce the expression of apicobasal and planar cell polarity genes in the tracheal stem cells which appear to be needed to ensure unidirectional migration. Finally, the authors suggest that trafficking and vesicular transport of Upd2 from the fat body towards the tracheal cells might be important.StrengthsThe manuscript is well written and presents extensive and varied experimental data to show a link between Upd2-JAK/STAT signaling from the fat body and tracheal progenitor cell migration. The authors provide convincing evidence that the fat body, located near the trachea, secretes vesicles containing the Upd2 cytokine and that affecting JAK-STAT signaling results in aberrant migration of some of the tracheal stem cells towards the anterior. Using ChIP-seq as well as analysis of GFP-protein trap lines of planar cell polarity genes in combination with RNAi experiments, the authors show that STAT92E likely regulates the transcription of planar cell polarity genes and some apicobasal cell polarity genes in tracheal stem cells which appear to be needed for unidirectional migration. The work presented here provides some novel insights into the mechanism that ensures polarized migration of tracheal stem cells, preventing bidirectional migration. This might have important implications for other types of directed cell migration in invertebrates or vertebrates including cancer cell migration. Overall, the authors have substantially improved their manuscript since the first submission but there are still some weaknesses.WeaknessesOverall, the manuscript lacks insights into the potential significance of the observed phenotypes and of the proposed new signaling model. Most of our concerns could be dealt with by adjusting the text (explaining some parts better and toning down some statements).(1) Directional migration of tracheal progenitors is only partially compromised, with some cells migrating anteriorly and others maintaining their posterior migration, a quite discrete phenotype.The strongest migration defects quantified in graphs (e.g. 100 μm) are not shown in images, since they would be out of frame, it would be beneficial to see them. In addition, the consequence of defects in polarized migration on tracheal development is not clear and data showing phenotypes on the final trachea morphology in pupae are not explained nor linked to the previous phenotypes.

We agree with you that it is informative to show strong anterior migration (> 100 μm). Accordingly, we have shown examples in Figure 3B and Figure 7R-S. In addition, we have also discuss on the links between migration defects and the consequential phenotypes of the animal at a later developmental stage in the revised manuscript. The undisciplined migration leads to insufficient regeneration and incomplete remodeling of airway and causes pupal lethality.

(2) Some important information is lacking, such as the origin of mutant and UAS-RNAi lines, which are not reported in the material and methods. For instance, mutants for components of the JAK-STAT pathway are used but not described. Are they all viable at the pupal stage? Otherwise, pupae would not be homozygous mutants. From the figure legend, it seems that the Stat92EF allele has been used, which is a point mutation, thus not leading to an absence of protein. If the hopTUM allele has been used, as mentioned in the legend, it is a gain-of-function allele. Thus, the authors should not conclude that "The aberrant anterior migration of tracheal progenitors in the absence of JAK/STAT components led to impairment of tracheal integrity and caused melanization in the trachea (Figure 3-figure supplement 1E-I)".

We apologize for inadequate description of the experimental materials and methods. We have listed the stock number of mutant and RNAi alleles in Key resource table and Materials. The mutant alleles that we chose to examine can survive to pupal stage, which is key to the success of our subsequent characterization of these mutants. According to your suggestion, we modified the statement for accuracy.

(3) The authors observe that tracheal progenitors display a polarized distribution of Fat that is controlled by JAK-STAT signaling. However, this conclusion is made from a single experiment using only 3 individuals with no statistics. This is insufficient to support the claim that "JAK/STAT signaling promotes the expression of genes involved in planar cell polarity leading to asymmetric localization of Fat in progenitor cells", as mentioned in the abstract, or that "the activated tracheal progenitors establish a disciplined migration through the asymmetrical distribution of polarity proteins which is directed by an Upd2-JAK/STAT signaling stemming from the remote organ of fat body."

We performed multiple biological replicates for Ft distribution experiments and observed similar trend, although we only showed three representative samples. In the revised text, we have included n number for statistic representation and statistic test.

(4) The authors demonstrate that Upd2 is transported through vesicles from the fat body to the tracheal progenitors. It remains somewhat unclear in the proposed model how Upd2 activates JAK-STAT signaling. Are vesicles internalized, as it seems to be proposed, and thus how does Upd2 activate JAK-STAT signaling intracellularly? Or is Upd2 released from vesicles to bind Dome extracellularly to activate the JAK-STAT pathway? Moreover, it is not clear nor discussed what would be the advantage of transporting the ligand in vesicles compared to classical ligand diffusion.

We do not know whether the association between Upd2 and Lbm is inside or outside vesicles. The vesicular trafficking of Upd2 is our observation and supported by various genetic and biochemical experiments. Our research does not imply the message that this vesicular trafficking has advantage over diffusion.